# Research

ecology, evolution

animal-mediated seed dispersal, coevolution, fruit coloration, plant–frugivore interaction, primatology, sensory adaptation

**Author for correspondence:**
Renske E. Onstein
e-mail: onsteinre@gmail.com

# Palm fruit colours are linked to the broad-scale distribution and diversification of primate colour vision systems

Renske E. Onstein[1,2], Daphne N. Vink[2], Jorin Veen[2], Christopher D. Barratt[1,3], Suzette G. A. Flantua[2,4], Serge A. Wich[2,5] and W. Daniel Kissling[2]

[1]German Centre for Integrative Biodiversity Research (iDiv) Halle-Jena-Leipzig, Deutscher Platz 5e, 04103 Leipzig, Germany
[2]Institute for Biodiversity and Ecosystem Dynamics (IBED), University of Amsterdam, PO Box 94240, Amsterdam, The Netherlands
[3]Department of Primatology, Max Planck Institute for Evolutionary Anthropology, Deutscher Platz 6, 04103 Leipzig, Germany
[4]Department of Biological Sciences, University of Bergen, PO Box 7803, 5020, Bergen, Norway
[5]School of Natural Sciences and Psychology, Liverpool John Moores University, Byrom Street, L33AF, Liverpool, UK

REO, 0000-0002-2295-3510

A long-standing hypothesis in ecology and evolution is that trichromatic colour vision (the ability to distinguish red from green) in frugivorous primates has evolved as an adaptation to detect conspicuous (reddish) fruits. This could provide a competitive advantage over dichromatic frugivores which cannot distinguish reddish colours from a background of green foliage. Here, we test whether the origin, distribution and diversity of trichromatic primates is positively associated with the availability of conspicuous palm fruits, i.e. keystone fruit resources for tropical frugivores. We combine global data of colour vision, distribution and phylogenetic data for more than 400 primate species with fruit colour data for more than 1700 palm species, and reveal that species richness of trichromatic primates increases with the proportion of palm species that have conspicuous fruits, especially in subtropical African forests. By contrast, species richness of trichromats in Asia and the Americas is not positively associated with conspicuous palm fruit colours. Macroevolutionary analyses further indicate rapid and synchronous radiations of trichromats and conspicuous palms on the African mainland starting 10 Ma. These results suggest that the distribution and diversification of African trichromatic primates is strongly linked to the relative availability of conspicuous (versus non-conspicuous) palm fruits, and that interactions between primates and palms are related to the coevolutionary dynamics of primate colour vision systems and palm fruit colours.

## 1. Introduction

The interaction between fruits and frugivores (i.e. fruit-eating and seed-dispersing animals) is prominent in tropical rainforests [1,2]. This mutualism relies on the intricate dependence of plants on frugivores to disperse their seeds, while frugivores are rewarded with important nutrients from the fruits they feed on [2]. Both fruits and frugivores have evolved adaptive traits to facilitate their interactions [2,3], and these traits may also influence co-diversification [4,5] and long-distance seed dispersal [6]. Frugivores and fruit plants are, therefore, able to shape each other's traits and the composition and structure of ecological communities [7]. Primates are important seed dispersers in tropical forests, particularly for large fruits (often larger than expected from primate gape widths), and fruit eating is facilitated by their teeth and hands [2,8]. In addition, they disperse seeds over

long distances and in multiple habitats owing to their relatively large home ranges [9]. The 'frugivory hypothesis' [10] states that primate colour vision systems coevolved with the morphological design of fruits (e.g. colour). Although this hypothesis has been tested experimentally for a few species and in specific study systems (e.g. [11]), it remains largely untested at broad macroecological and macroevolutionary scales.

Trichromatic colour vision is the ability to distinguish red from green colours. Fruits with bright colours such as yellow, orange and red are thought to appear 'conspicuous' to trichromatic primates, especially when foraging against a background of green foliage [11–13]. Trichromatic vision could thus provide a competitive advantage over frugivores that cannot distinguish reddish colours from green (e.g. 'dichromatic' primates). Among placental mammals, trichromatic vision is unique in primates (including humans). Interestingly, primates show variations in trichromatic colour vision systems. Some species have 'polymorphic' colour vision (primarily platyrrhine monkeys), in which part of the population is dichromatic and part is trichromatic, whereas other species are 'routine trichromatic' (primarily African and Asian monkeys and apes) in which all individuals have trichromatic vision [14]. This variation may in part explain why the ecological factors that have driven the evolution of colour vision systems in primates remain debated [12,13]. In addition, besides fruit detection, trichromacy has been suggested to be an adaptation to detect predators [15] and young leaves [16].

It is posited that if 'frugivory' has been the selective pressure for the evolution of trichromacy, then predominantly day-active, frugivorous primates should benefit from trichromatic vision [2,12]. Similarly, the advantage of trichromatic vision may then be enhanced in places and during periods of fruit scarcity when competition is high [16,17]. During times of fruit scarcity, 'keystone' plant resources, which have large crop sizes and an asynchronous fruiting pattern throughout the year, such as fruits of palms (Arecaceae) or figs (*Ficus*, Moraceae), are the primary food source for many frugivores (including primates) [2,18,19]. Nevertheless, it remains unclear whether conspicuous fruits and trichromatic vision have evolved contemporaneously, and to what extent co-diversification between keystone plant species with conspicuous fruits and trichromatic primates has influenced their broad-scale distribution and diversity (but see [16]).

Here, we integrate macroecological and macroevolutionary approaches to test the hypothesis that the origin, distribution and diversity of trichromatic vision in primates is positively associated with the availability of conspicuous fruits of palms (Arecaceae). Palms are keystone resources for frugivores and nearly all species are animal dispersed (mainly by birds, primates, bats and other mammals) [20]. Furthermore, there are numerous examples of palm–primate seed dispersal interactions [21–26]. First, we predict that assemblages dominated by palm species with conspicuous fruits also show the highest trichromatic and/or polymorphic primate species richness, both globally (P1) and in major biogeographic realms (P2), even when accounting for other confounding factors such as climate and forest height [27,28]. We also predict (P3) that the effect of conspicuous palm fruits on trichromatic and/or polymorphic richness peaks in regions with subtropical, seasonal climates, where food is likely to be a limiting factor and competition for resources high, but fruits are still an important food resource in the diet (e.g. as compared to primates in arid

regions [29,30]). Finally (P4), we predict that evolutionary radiations of trichromatic and/or polymorphic primates and palms with conspicuous fruits are synchronized, with rapid diversity increases occurring in parallel [4,31]. If true, this would provide macroevolutionary evidence for co-diversification of both interaction partners (*sensu* the 'primate/angiosperm coevolution theory' [32]).

## 2. Results

### (a) Global relationship between palm fruit colour and trichromatic primate species richness

To test whether palm assemblages dominated by species with conspicuous fruits are particularly diverse in trichromatic and/or polymorphic primates (P1), we quantified species richness of primates with these colour vision systems and the proportion of palms with conspicuous fruits within botanical countries worldwide (figure 1; see summary statistics in the electronic supplementary material, table S1). Globally, routine trichromatic primate richness is highest in countries with a high proportion of palm species that have conspicuous fruits (mainly African countries, with more than 67% of palm floras being dominated by palm species with conspicuous fruits, e.g. in Congo-Brazzaville, Ivory Coast, Equatorial Guinea and Liberia), whereas polymorphic vision dominates in the Americas where palm floras are predominantly non-conspicuous (e.g. 66% of palm floras in Costa Rica and southern Brazil are dominated by palms with non-conspicuous fruits, 90% in Guatemala) (figure 1). Using structural equation models (SEMs), we further tested whether (geographical) trichromatic and/or polymorphic primate richness is positively associated with the proportion of palm species that have conspicuous fruits, correcting for the direct and indirect effects of climatic variables, forest canopy height and area (see the electronic supplementary material, figure S1 for the base model). An SEM at the global scale supported P1 by showing a strong positive effect of the proportion of conspicuous palm fruits on day-active, frugivorous routine trichromat richness (std. coeff. = 0.554; figure 2b) and significant but weaker effects on polymorphic richness (std. coeff. = 0.171; figure 2a). In addition, the global SEM indicated that larger areas and less seasonal climates in temperature (but not precipitation) have higher trichromatic and/or polymorphic richness (figure 2a,b; electronic supplementary material, figure S2). Importantly, when comparing the observed effect of the proportion of conspicuous palm fruits on trichromatic and/or polymorphic richness to 1000 simulated effects (in which we randomly distributed primate species across botanical countries), only the effect on routine trichromatic primates remained significant (i.e. outside the 95% quantile of simulated effects). By contrast, the effect on polymorphic richness did not deviate from a random expectation (electronic supplementary material, figure S3). Analyses focusing on all primates versus only day-active, frugivorous species provided qualitatively similar results (electronic supplementary material, figures S2 and S3).

We performed additional sensitivity analyses to assess whether the positive association between trichromacy and conspicuous palm fruits could result from confounding correlations between palm fruits and other variables [15,16]. Indeed, when implementing a similar SEM with non-frugivorous trichromatic and/or polymorphic primate richness

(a) global polymorphic primate species richness

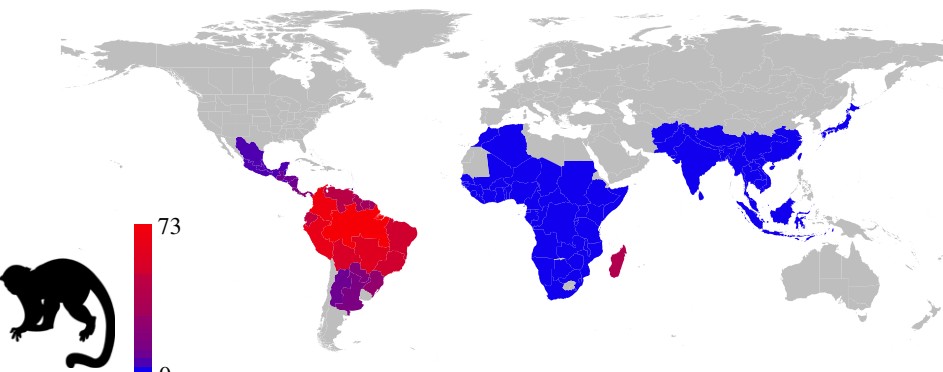

(b) global trichromatic primate species richness

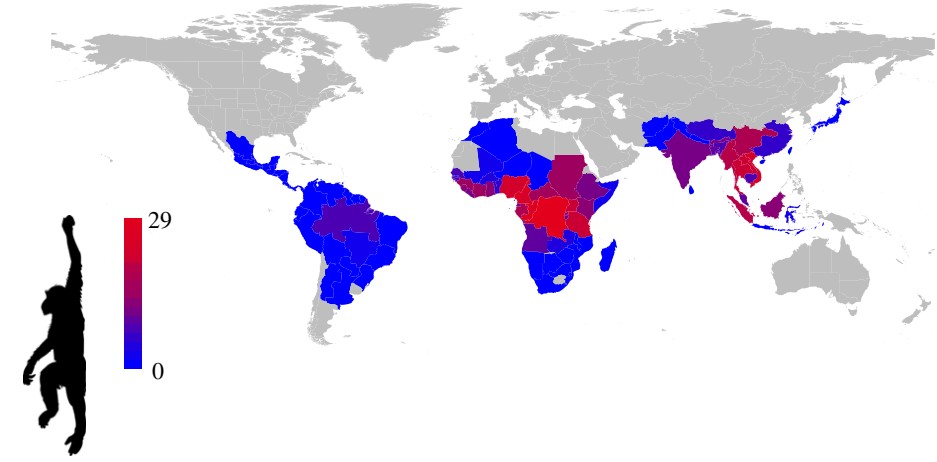

(c) global proportion of conspicuous palm fruits

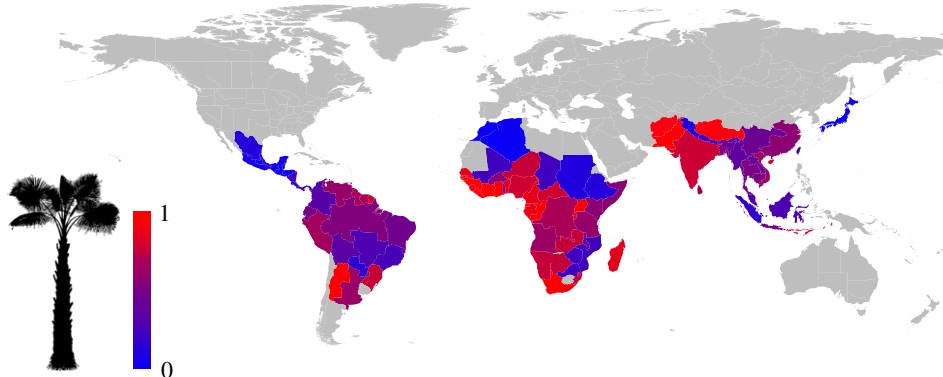

**Figure 1.** Global variation in trichromatic colour vision of primates and conspicuous fruit colours of palms. Maps depict the global distribution and variation of: (a) polymorphic primates, (b) routine trichromatic primates, and (c) the proportion of palms with conspicuous (i.e. yellow, red or orange) fruit colours (from a total including all palm species with primate fruits, i.e. brown, green, orange, yellow, red and purple fruits). Values are quantified at the spatial resolution of botanical countries. Grey colours indicate countries where no primate and palm species occur. (Online version in colour.)

as the response variable, conspicuous palm fruits failed to explain any of the variation (electronic supplementary material, table S2). Similarly, the proportion of conspicuous palm fruits did not explain dichromatic primate richness (electronic supplementary material, table S2). Last, results were robust with respect to continental differences in total routine trichromat richness (Americas $n = 14$, Africa $n = 62$, Asia $n = 82$), but not with respect to polymorphic richness (Americas $n = 113$, Africa $n = 14$, Asia $n = 0$) (electronic supplementary material, table S2).

## (b) Biogeographic differences in the relationship between palm fruit colour and trichromatic primate species richness

We applied SEMs separately for three biogeographic realms (Africa, Americas and Asia) to evaluate biogeographic differences in the effect of the proportion of conspicuous palm fruits on trichromatic and/or polymorphic richness (P2). Consistent with the global result, we detected a positive relationship in Africa, both when focusing on routine trichromatic primates

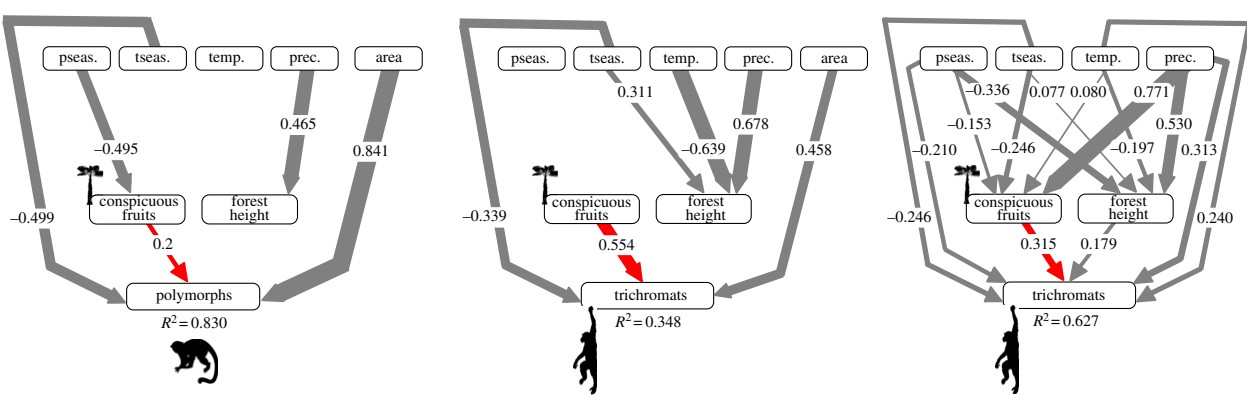

**Figure 2.** Structural equation models (SEMs) showing the effects of predictor variables on day-active, frugivorous polymorphic and trichromatic primate species richness. SEMs illustrate how the proportion of conspicuous palm fruits (red arrows) determines (*a*) polymorphic and (*b*,*c*) trichromatic primate richness at (*a*,*b*) the global scale (botanical country resolution) and (*c*) across mainland Africa (where no polymorphic primates occur) (resolution of 110 × 110 km grid cells) once direct and indirect effects of forest height, climate and area are accounted for. Effect sizes represent standardized coefficients and arrows indicate the direction of the effect, with arrow thickness being proportional to effect strength. Only statistically significant effects (at $p < 0.05$) are illustrated. Results are qualitatively similar to those obtained focusing on all (not only day-active, frugivorous) primates (electronic supplementary material, figures S2 and S7). prec. = annual precipitation, temp. = annual mean temperature, tseas. = temperature seasonality, pseas. = precipitation seasonality, conspicuous fruits = proportion of conspicuous palm fruits, polymorphs = day-active, frugivorous, polymorphic primate species richness, trichromats = day-active, frugivorous, routine trichromatic primate species richness. (Online version in colour.)

only (std. coeff. = 0.289, excluding Madagascar where no trichromatic primates occur) or on polymorphic/trichromatic primates combined (std. coeff. = 0.178, including Madagascar), but only when restricting the dataset to only include day-active frugivorous primates (electronic supplementary material, figure S4). By contrast, in the Americas and Asia (electronic supplementary material, figures S5 and S6) no statistically significant relationship between primate richness and proportion of conspicuous palm fruits was found, with the exception of a negative effect of conspicuous palm fruits on routine trichromatic primates (i.e. howler monkeys) in the Americas (std. coeff. = −0.577). Hence, only the African assemblages resembled the global signal of palm fruit colour on trichromatic primate richness.

An in-depth analysis across mainland Africa (where polymorphic primates are absent) using palm distribution data at a finer spatial resolution (i.e. with grid cells of 110 × 110 km) also showed a positive effect of conspicuous palm fruits on trichromat richness (std. coeff. = 0.193, electronic supplementary material, figure S7) and the strength of this effect increased when only including day-active, frugivorous primates (std. coeff. = 0.315; figure 2*c*). This analysis also revealed positive but weaker effects of forest height, temperature and precipitation, and negative effects of temperature seasonality and precipitation seasonality on trichromat richness (figure 2*c*; electronic supplementary material, figure S7). These results were robust with respect to randomisations of primate occurrences (electronic supplementary material, figure S8), and indicate that the highest richness of trichromatic primates in mainland Africa is found in warm and wet areas with low seasonality, and in places with tall forest canopies as well as high proportions of conspicuous palm fruits.

Repeating the African (grid-based) SEM analyses by also including all 68 African fig species (*Ficus*, Moraceae)—another important keystone plant resource for primates [33]—showed that the effect of conspicuous fruits on day-active, frugivorous trichromat richness was still significant but weaker than for palms only (std. coeff. = 0.150; electronic supplementary material, table S2). A weaker relationship was also apparent

when correcting for spatial autocorrelation (see Supplementary methods in the electronic supplementary material). Consistent with the main result, we also detected a positive effect of species richness of palms with conspicuous fruits (rather than using the proportion of conspicuous fruits) on African trichromat richness (std. coeff. = 0.275; electronic supplementary material, table S2), whereas the effect of species richness of palms with non-conspicuous fruits on trichromat richness was not supported (as expected; electronic supplementary material, table S2).

### (c) The strongest effect of conspicuous palm fruits on routine trichromatic primate species richness is in the subtropics

To test whether the effect of conspicuous palm fruits on trichromat richness peaks in regions with subtropical climates (P3), we repeated the African (grid-based) SEM analysis by progressively removing grid cells based on the minimum number of palm species present. Consistent with the expectation, we found that the standardized coefficients of the proportion of conspicuous palm fruits on day-active, frugivorous trichromat richness (obtained from SEMs) initially increased along a gradient from areas with low food plant diversity to areas with high diversity (i.e. from arid to subtropical regions). Towards the most species-rich places (i.e. tropical rainforests where the species richness of conspicuous palms is also high), the standardized coefficients again decreased, following a hump-shaped relationship (figure 3; see the electronic supplementary material, figure S9 when also including figs and when focusing on all primates instead of day-active frugivores only). This result suggests that the strongest effect of conspicuous palm fruits on trichromat richness is found in regions with low-to-intermediate levels of palm food plant species richness, i.e. in arid-subtropical transition zones where competition for fruits may be high (see the electronic supplementary material, figure S10 for the spatial

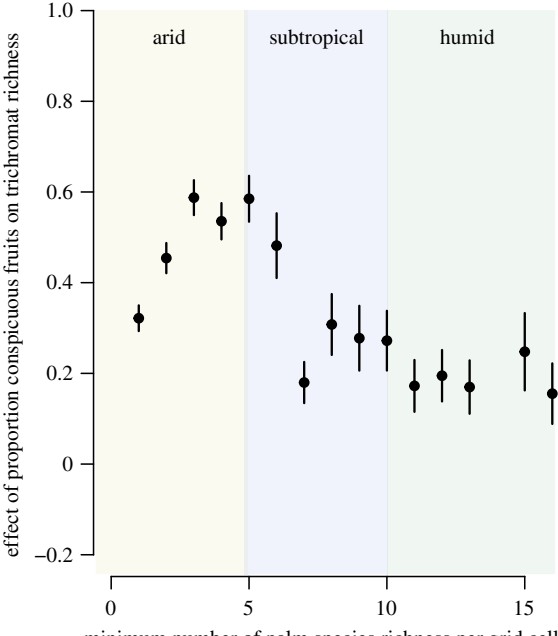

**Figure 3.** The importance of the proportion of conspicuous palm fruits on day-active, frugivorous, trichromatic primate species richness along a gradient from arid to humid tropical climates in mainland Africa. Standardized coefficients (±s.d.) representing the effect of the proportion of conspicuous palm fruits on day-active, frugivorous, trichromatic primate species richness were obtained from structural equation models (SEMs) ($n = 16$). These SEMs were fitted by progressively delimiting the number of included grid cells based on the minimum number of palm food plant species richness present (see the electronic supplementary material, figure S10 for spatial delimitation). Effect strength peaks in arid-subtropical transition zones with low-to-intermediate levels of palm food plant species richness. (Online version in colour.)

delimitation of regions and climate zones based on palm food plant species richness).

### (d) Synchronized evolutionary radiations of routine trichromats and palms with conspicuous fruits in mainland Africa

To evaluate whether evolutionary radiations of polymorphic and/or routine trichromatic primates and palms with conspicuous fruits are synchronized (P4), we implemented ancestral state reconstructions and assessed diversity increases in polymorphic and trichromatic primates and palms with conspicuous fruits. Although polymorphic vision evolved contemporaneously with conspicuous palm fruits at a global scale ca 30–40 Ma, the evolution of routine trichromatic vision happened later, ca 20–30 Ma (electronic supplementary material, figure S11). Nevertheless, routine trichromats showed rapid diversity increases from 10 Ma onwards at a global scale (figure 4a) as well as within Africa (figure 4e) and Asia (figure 4g), and in parallel with palms that have either conspicuous or non-conspicuous fruits (figures 4b,d,f,h). Notably, African palms with conspicuous fruits (e.g. in genera Eremospatha, Laccosperma) increased in diversity slightly more rapidly than those with non-conspicuous fruits from 10 Ma onwards (figure 4f), with a similar stronger diversity increase of mainland African trichromats compared to primates with no trichromatic colour vision (figure 4e, but see the electronic supplementary material, figure S12 when including the rapid

radiation of dichromatic lemurs on Madagascar). Similar to palms, African figs with conspicuous fruits (as compared to those with non-conspicuous fruits) also showed rapid diversity increases from 10 Ma onwards (electronic supplementary material, figure S13). Faster diversification of primates with trichromatic colour vision (compared to polymorphic or monochromatic/dichromatic primates) and of palms with conspicuous fruits (compared to palms with non-conspicuous fruits) was also supported by an additional analysis using a multiple state speciation and extinction (MuSSE) model (electronic supplementary material, figure S14). These results support the prediction (P4) that rapid diversity increases of trichromats have occurred in parallel with palms, but only on the African mainland. This contrasts with routine trichromats in the Americas which maintained a relatively low diversity through time, whereas polymorphic primates in the Americas showed rapid diversity increases from 10 Ma onwards (figure 4c).

## 3. Discussion

### (a) The macroecological drivers of routine trichromatic primate richness

Our results show that at a global scale, the geographical distribution of trichromatic primates is associated with the availability of conspicuous relative to non-conspicuous palm fruits (figures 1 and 2). However, when including polymorphic primates, our results do not deviate from a neutral expectation in which primates are randomly distributed across regions. This may be explained by the fact that colour vision systems evolved only few times independently within primates (probably less than seven times, electronic supplementary material, figure S11), and colour vision is thus geographically (figure 1) and phylogenetically (electronic supplementary material, figure S11) constrained. Our results are, therefore, consistent with the frugivory hypothesis [10] for routine trichromacy, especially in Africa (figure 2; electronic supplementary material, figures S7 and S8), but the ecological determinants of polymorphic vision remain questionable.

The positive relationship between the species richness of trichromats and the distribution of palm fruits with conspicuous colours is consistent with field studies which demonstrate that fruit colours are related to the colour vision of local seed disperser communities. For instance, studies that compare fleshy fruits from mainland Africa (where trichromatic day-active primates occur) with those from Madagascar (where lemurs are primarily night-active and red–green colour-blind) suggest that fruit colours of plant communities in mainland Africa are more conspicuous than those in Madagascar [7,34]. Detailed dietary studies of some trichromatic frugivorous primates in Africa such as the Tana River Mangabey (Cercocebus galeritus) provide further evidence that conspicuous palm fruits alone can make up to 20% of their total diet [23].

### (b) Trichromacy and palm fruit colours: an African phenomenon

Interestingly, the macroecological relationship between polymorphic and/or trichromat richness and conspicuous fruits was not evident in the Americas and Asia. Indeed, most primates in the Americas have polymorphic colour vision and

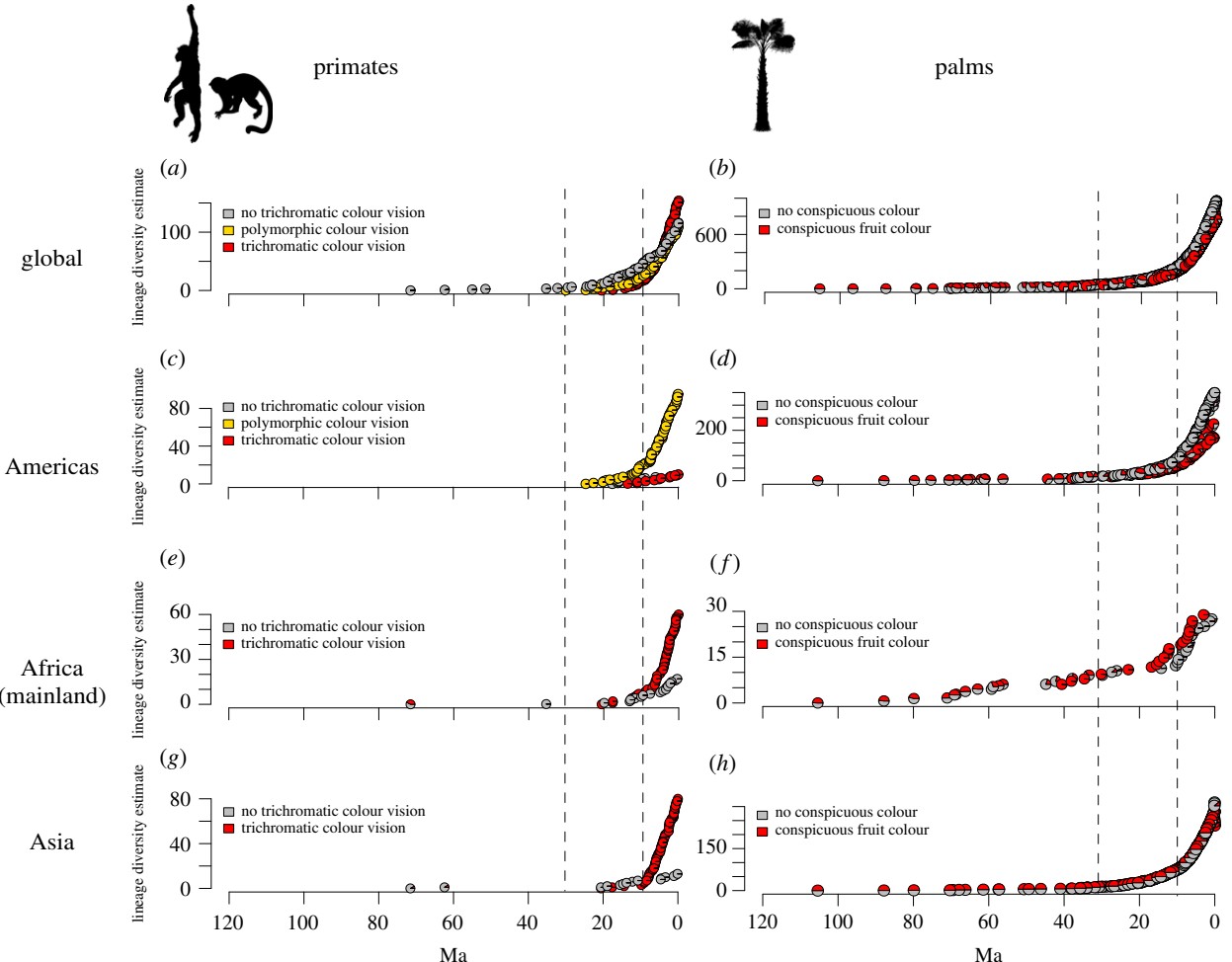

**Figure 4.** Diversity changes of trichromatic and polymorphic primates and conspicuous palm fruits through geological time. The pie charts reflect ancestral nodes in the phylogenies and their probability to have the focal trait, obtained from ancestral state reconstructions: trichromatic (red) or polymorphic (yellow) vision in primates (a,c, e,g) and conspicuous fruit colours (red) in palms (b,d,f,h). Globally, trichromatic primates (a) and palms (b) show rapid diversity increases at *ca* 10 Ma, after the innovation of polymorphic and routine trichromatic vision *ca* 20–30 Ma. In the Americas, (c) polymorphic primates (yellow colour in pie charts) and (d) non-conspicuous palm fruits (grey colour in pie charts) show rapid diversity increases from 10 Ma onwards. In Africa (excluding Madagascar), parallel radiations of trichromatic primates (e) and conspicuous palms (f) are initiated *ca* 20 Ma, with rapid diversity increases from *ca* 10 Ma onwards, compared to more gradual diversification of non-trichromatic African primates, and lineages with non-conspicuous palm fruits. Asia also shows rapid increases of trichromatic primates (g) from 10 Ma onwards and both palm lineages with conspicuous or non-conspicuous fruits (h) diversify rapidly during this time. (Online version in colour.)

prefer palm fruits with non-conspicuous colours (e.g. titi monkeys in the genera *Callicebus*, *Plecturocebus* or *Cheracebus*) [16,21,22]. Furthermore, all routine trichromats in the Americas (howler monkeys) are predominantly folivorous [35] and have digestive tracts adapted to fermenting leaves [36]. By contrast, many trichromatic primates in Asia (e.g. macaques) feeding on palms have an overall highly frugivorous diet [35], but the colours of palm fruits that they feed on (e.g. species in genera *Caryota* and *Calamus*) are both non-conspicuous and conspicuous [25]. This flexibility could at least partly explain why the relationship between trichromat richness and proportion of conspicuous palm fruits in Asia was not statistically significant. Selective agents other than primates, such as birds and bats, may also act on fruit colours [13,37], especially in the Americas and Asia where birds may be more important seed dispersers than primates, relative to Africa which is comparably poor in frugivorous birds [38]. This provides an alternative explanation for the absence of a relationship between conspicuous fruits and trichromacy outside Africa.

Trichromatic colour vision in Africa has also been hypothesized to be adaptive to detecting young, reddish leaves [16]. Dominy *et al.* [16] argued that increased seasonality after the

onset of the Oligocene–Miocene led to an overall reduction in keystone plant species, including palms [39,40], and an increase in periods of fruit scarcity. A fallback diet of young, reddish leaves may thus have been the selective pressure for the evolution of routine trichromatic vision (from a polymorphic ancestor, electronic supplementary material, figure S11) in these catarrhine primates. To test this competing hypothesis, we could use a similar SEM approach as done here for palm fruit colours, but then with detailed spatial and phylogenetic data of leaf coloration. These data are currently unavailable. However, results from our sensitivity analyses suggest that trichromacy is not simply correlated to conspicuous palm fruits owing to confounding correlations between fruits and other variables, such as reddish leaves (electronic supplementary material, table S2).

## (c) Palm fruit colour effects on trichromats in arid and subtropical African forests

Interestingly, even though total species richness of African palms and primates is highest in tropical regions, we found the strongest effect of conspicuous palm fruits on trichromat

richness in transition zones between arid and subtropical (African) forests (figure 3; electronic supplementary material, figure S10). These forests are home to several small-ranged primate species (e.g. *Chlorocebus djamdjamensis* and *Cercocebus sanjei*) which favour fruits especially in 'stressful' (e.g. fragmented or disturbed) areas [30] where competition for food is high and trichromacy may thus provide a competitive advantage. In line with this, chimpanzees, guenons and mangabeys rely heavily on the (conspicuous) fruits of the oil palm (*Elaeis guineensis*) and the date palm (*Phoenix reclinata*) during the dry seasons in Gabon and Kenya, respectively, as these palm species fruit asynchronously, i.e. with a few fruits ripening at a given time but over long periods [24,26].

## (d) A co-diversification scenario between primates and palms in Africa

The 'stressful' changes in the ecological settings on the African continent linked to cooling and drying in the Miocene [41,42], may also have increased primate competition for, and dependence on, keystone fruits. Consequently, trichromatic colour vision may have provided a competitive advantage to increase (ripe) fruit detection and intake, thus further increasing selection pressures on conspicuous palm fruit colours. This may have influenced interspecific trait divergence and primate/plant co-diversification, as facultative mutualism may lead to strong selection on coevolving traits and consequently increase diversification rates [4,5,32]. Such a scenario would be consistent with the rapid accumulation of diversities through time in both frugivores (e.g. trichromats) and food plants (e.g. conspicuous palms and figs) (figure 4; electronic supplementary material, figure S13). Indeed, although diffuse coevolution between primates and angiosperms may date back to the Early Cenozoic (*ca* 66 Ma) [32], our results suggest that, at least in Africa, mutualism-dependent diversification between frugivorous trichromatic primates and conspicuous palms may have happened from *ca* 10 Ma onwards (Middle to Late Miocene) (figure 4). This contrasts with folivorous primates which did not diversify faster or slower than non-folivorous primates in Africa from *ca* 10 Ma onwards (electronic supplementary material, figure S15). Furthermore, our results suggest that trichromats have also influenced the radiation of conspicuous (versus non-conspicuous) palms. The absence of a relationship between the diversification of palms and polymorphic primates suggests that polymorphs may not shape fruit colours in the same way as trichromats do, and fruit traits other than colour (e.g. husk thickness, fruit size, fruit odour, nutritional content and the presence of juicy soft pulp) may play a more important role in the selection of fruits by polymorphs.

A co-diversification scenario can be circumstantial and we cannot exclude the possibility that the radiation of African catarrhine primates was related to other (derived) catarrhine traits, such as their enlarged brains and complex social structures [43]. Hence, routine trichromacy could also have evolved in concordance with such other traits. Furthermore, Miocene climate change and Pleistocene glacial-interglacial fluctuations may have facilitated diversification through fragmentation, regardless of the ecological interactions between species. Co-phylogenetic methods could be used to test for true co-diversification of primates and food plants [44], but these methods require detailed interaction data between all species in both phylogenetic trees, which is currently not available for primates and their food plants.

## 4. Conclusion

Our study provides novel evidence that the geographical distribution and diversity of primates with trichromatic colour vision in Africa is associated with the broad-scale distribution of conspicuous fruits of palms. The macroecological and macroevolutionary analyses performed in this study do not prove causality, and additional evidence is needed to show that these co-occurrence patterns of palm fruit colours and primate species richness are related to actual seed dispersal interactions and therefore reflect ecological impact. Currently, many trichromatic primates are threatened with extinction [45] owing to rapid environmental degradation and global change. This may have cascading effects on their seed dispersal services, especially when fruit detection relies on the visual capabilities of local seed disperser communities. The conservation of plant–animal interactions, fruit colour diversity and consumer colour vision systems is thus essential for the maintenance of tropical biodiversity.

## 5. Material and methods

### (a) Primate data

From a total of 411 primates (following the International Union for Conservation of Nature (IUCN) species classification), 158 primates (38% of total) have routine trichromatic colour vision (i.e. both males and females have trichromatic colour vision) and 126 primates (31% of total) have polymorphic vision (i.e. part of the population is dichromatic and part is trichromatic). Knowledge on trichromatic colour vision and activity status (day-active/night-active/crepuscular) was derived from the literature (electronic supplementary material, Appendix S1). Frugivores were defined as species having fruits as at least some part of their diet (using rank 1–3 of the 'Fruit' category from the MammalDIET dataset [46]). For more details on data collection see Supplementary methods in the electronic supplementary material. All trait data and references are available from the electronic supplementary material, Appendix S1 and the Dryad Digital Repository: https://doi.org/10.5061/dryad.6hdr7sqwn [47].

To quantify the global distribution of trichromatic and polymorphic primate species richness, we used primate distribution data (geographical range maps) from the IUCN Global Mammal Assessment [45]. These maps were intersected with a polygon file of 'botanical countries', i.e. standardized sampling units as defined by the International Working Group on Taxonomic Databases for Plant Sciences (TDWG) [48] for which palm distribution data are available (see below). For the African analysis, we also created a second dataset with a finer spatial resolution by intersecting the primate range maps with an equal-area grid of $110 \times 110$ km. Trichromatic and/or polymorphic species richness was derived as the sum of all those species being present in each botanical country (global dataset) or each $110 \times 110$ km grid cell (Africa dataset). To obtain a phylogeny for extant primates, we pruned the mammal phylogenetic tree from [49] to include our taxa of interest ($n = 385$ primate species, i.e. 26 primate species were missing from the phylogeny).

### (b) Palm data

From a total of 2557 palm species (following the World Checklist of palms [50]), ripe fruit colour data were assembled for 1749 species (*ca* 70% of total). Following [16], we classified orange, red, yellow

and pink fruits as 'conspicuous' (i.e. light reflectance spectra of fruits and leaves differ [12]), and brown, black, green, blue, cream, grey, ivory, straw-coloured, white and purple fruits as 'non-conspicuous'. We confirmed the classification of conspicuous palm fruits based on human vision by measuring fruit reflectance spectra from 54 fresh palm fruits belonging to 18 species (for details see Supplementary methods, table S3 and figure S16 in the electronic supplementary material). As primates mostly feed on brown, green, orange, yellow, red and purple fruits [2], we excluded palm fruits that did not have these colours in their description from the analyses ($n = 1444$ palm species remained). For more details on data collection see Supplementary methods in the electronic supplementary material. All trait data, criteria and references are available from the electronic supplementary material, Appendix S2 and [47].

At the global scale, we summed up palm presence–absence data (only including species with brown, green, orange, yellow, red and purple fruits) at the spatial resolution of botanical countries, and quantified species richness of palms with conspicuous and non-conspicuous fruits (electronic supplementary material, figure S17). For the African analysis, we used palm species distribution maps and intersected them with an equal-area grid of $110 \times 110$ km cell size. We calculated the proportion of palms with conspicuous fruit colours in each grid cell by dividing the total number of conspicuous palm species by the sum of all non-conspicuous and conspicuous palm species. Palm distribution maps were based on species distribution models using 5500 occurrence records representing 62 (out of 65) African palm species (for details, see [40]). For the palm phylogeny, we used the maximum clade credibility phylogenetic tree from [5] ($n = 1742$ palm species, i.e. seven palm species with trait data were missing from the phylogeny).

## (c) Environmental data

In addition to palm fruit colour, environmental factors may also influence the distribution and diversity of (trichromatic) primates [27,28]. We, therefore, included several environmental variables in our statistical analyses. We selected four climatic variables from the WorldClim database that were not strongly correlated (Spearman's rank $r < 0.7$) and which represent climatic factors that are important for the growth and survival (and thus species richness) of plants and animals. These variables were annual mean temperature (°C*10), annual precipitation (mm), temperature seasonality (standard deviation °C*10) and precipitation seasonality (coefficient of variation in millimetres). Additionally, as a proxy for forest height and vertical forest structure [28] we used remotely sensed canopy height data available globally at 500 m resolution [51]. For all environmental variables, we calculated average values per botanical country (global analysis) or $110 \times 110$ km grid cell (Africa analysis). Furthermore, the area of each botanical country (calculated in ESRI ArcGIS with a Cylindrical Equal Area projection) was included in the global analysis as a covariate because TDWG units are unequal in size and larger areas generally support higher species richness. The area was not necessary for the African analysis because all grid cells were of equal area.

## (d) Structural equation modelling

We used SEMs to test whether spatial variation in trichromatic and/or polymorphic species richness is positively related to the proportion of palm species with conspicuous fruit colours (P1, P2). We included climatic variables, canopy height, area and the proportion of conspicuous palm fruits in the models and normalized all variables between 0 and 1. We used a sqrt-transformation to obtain normality in trichromat and/or polymorphic richness and a log-transformation for area. Only botanical countries with more than zero trichromats and/or polymorphs and more than two palm species were included in the analyses to avoid erroneous

estimates of the proportion of conspicuous palms (resulting in 93 botanical countries globally).

We started our modelling process with an *a priori* SEM that included all hypothesized pathways among all predictor variables (electronic supplementary material, figure S1) and evaluated the model's modification indices, model fits and residual correlations [52]. To ensure an adequate fit of SEMs, we made sure that *p*-values of $\chi^2$-tests $> 0.05$, comparative fit index (CFI) $> 0.90$ and confidence intervals of the root mean square error of approximation (RMSEA) $< 0.05$ [52]. We progressively deleted paths with the least statistical significance from the SEM until our final SEM only consisted of significant pathways (at $p < 0.05$), for which we extracted the standardized coefficients. The model residuals were tested for normality and equal variances, and we checked for extreme outliers that could affect the results.

We repeated the global SEM procedure separately for the three biogeographic realms (Africa, Americas and Asia). For Africa we also repeated the SEM procedure using the grid cell dataset, thereby increasing the sample size ($n = 794$ grid cells) and the spatial resolution ($110 \times 110$ km) of the dataset. Furthermore, all SEMs were performed for trichromatic, polymorphic and trichromatic/polymorphic primates combined. In addition, we repeated the SEMs including only day-active, frugivorous primates because we expected those to be more strongly associated with the (relative) availability of conspicuous palm fruits. For a complete overview of all analyses and results, see the electronic supplementary material, table S4.

Last, we repeated the African SEM analysis by progressively removing grid cells from the analysis based on the minimum number of palm food plant species, to test whether the effect of conspicuous palm fruits on trichromat richness derived from SEM peaks in regions with subtropical climates (P3). By including grid cells with a minimum of 2 to a minimum of 16 palm species (i.e. a total of 15 SEMs), we illustrate the progressive spatial delimitation of subtropical regions, eventually only including the most species-rich places (e.g. tropical rainforests with greater than 10 palm species per grid cell) (electronic supplementary material, figure S10). We examined the standardized coefficients of conspicuous palm fruits on trichromat richness in these SEMs. The SEMs were calculated using the R package 'lavaan' [53]. For more details on the approach, see Supplementary methods in the electronic supplementary material.

## (e) Simulations and sensitivity analyses

To evaluate whether an association between colour vision and palm fruit colour could result from confounding correlations, we performed simulations and six sensitivity analyses. For the simulations, we randomly reshuffled primate species and their colour vision across botanical countries, keeping the total species richness per botanical country constant, and then repeated the SEMs and extracted the standardized coefficients of the proportion of conspicuous palm fruits on primate richness. We repeated this 1000 times. We compared whether our observed effect was higher than 95% of the simulated effects, which would support the idea that the distribution of primate colour vision deviates from a random expectation. We also repeated these simulations for the African grid-based analysis. For the sensitivity analyses, we repeated the SEMs but (i) used the diversity of non-frugivorous trichromats or (ii) dichromat species richness as the response variable; (iii) corrected for total trichromatic/polymorphic richness differences between biogeographic realms (Americas $n = 14/113$, Africa $n = 62/14$, Asia $n = 82/0$); (iv) followed a more conservative classification of frugivory [35]; (v) used conspicuous or non-conspicuous palm fruit species richness (instead of the proportion) to explain primate richness; and (vi) evaluated if results were consistent with another keystone food plant clade, namely figs (*Ficus*, Moraceae) [33]. We show that our results are robust with respect to

type I error rates (rejecting a false null hypothesis), species richness differences between continents, the classification of diet, using the proportion rather than the absolute richness of palms that have conspicuous fruits (but only in mainland Africa), and including figs. For more details on the sensitivity analyses see Supplementary methods and table S2 in the electronic supplementary material.

## (f) Spatial autocorrelation

Spatial autocorrelation can affect results from non-spatial analyses [54]. We, therefore, additionally fitted simultaneous autoregressive (SAR) models, which allow us to account for spatial autocorrelation in model residuals [54]. We first fitted non-spatial, ordinary least-squares (OLS) regression models with the same set of predictor variables on trichromatic and/or polymorphic richness as in the SEMs, and then fitted similar SAR models (for details see Supplementary methods in the electronic supplementary material). Because standardized coefficients of the SAR models were similar to those of the OLS models and SEMs, we only present the standardized coefficients from the SEMs in the main text.

## (g) Ancestral colour vision and palm fruit colour reconstructions

To assess whether evolutionary radiations of trichromatic and/or polymorphic primates and palms with conspicuous fruits are synchronized (P4), we visualized the evolution of trichromatic and polymorphic colour vision and conspicuous palm fruits on the primate and palm phylogenetic trees, respectively (electronic supplementary material, figure S11). We used the marginal probabilities of ancestral nodes reconstructed as trichromatic or polymorphic colour vision (in primates) or conspicuous fruit colours (in palms) to estimate the lineage diversity present with those traits at the time of occurrence of those particular nodes (for more details,

see [55]). To assess whether the global pattern is consistent across biogeographic realms, we repeated these analyses separately for the Americas, Africa and Asia. Analyses were performed using the 'make.simmap' and 'estDiversity' functions in the 'phytools' R package [56]. For more details, see Supplementary methods in the electronic supplementary material.

Data accessibility. Functional trait data of primates, palms and figs are available from Appendix S1, Appendix S2 and Dryad Digital Repository: https://doi.org/10.5061/dryad.6hdr7sqwn [47]. Geographical data of botanical countries with trichromatic and polymorphic primate species richness, mean climate variables and proportions of conspicuous/non-conspicuous palm fruits are available at [47].

Authors' contributions. R.E.O. and W.D.K. designed the research with contributions from all authors; R.E.O., W.D.K., D.N.V., J.V. and C.D.B. collected data; R.E.O. and W.D.K. performed the analyses; C.D.B. and S.G.A.F. helped with the spatial aggregation of datasets; R.E.O. wrote the manuscript with major contributions from W.D.K. and input from all authors.

Competing interests. The authors declare no competing financial interests.

Funding. W.D.K. acknowledges funding from the Netherlands Organization for Scientific Research (grant no. 824.15.007) and the University of Amsterdam (via a starting grant and through the Faculty Research Cluster 'Global Ecology'); R.E.O. and C.D.B. acknowledge the support of the German Centre for Integrative Biodiversity Research (iDiv) Halle-Jena-Leipzig funded by the Deutsche Forschungsgemeinschaft (DFG, German Research Foundation)—FZT 118. R.E.O. further acknowledges SYNTHESYS grant no. GB-TAF-6695.

Acknowledgements. We thank Xishuangbanna Tropical Botanical Garden (China), the 'Biogeography and Ecology Group' and Yaowu Xing for assistance with palm collections and palm fruit colour measurements. Furthermore, we acknowledge the Royal Botanic Gardens Kew Herbarium and William Baker for using specimens and assistance during measurements. We thank Amanda Melin and two anonymous reviewers for helpful suggestions that improved the manuscript.

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
