## [Reviewer comments · Proceedings of the Royal Society B: Biological Sciences]

Review History

RSPB-2019-2114.R0 (Original submission)

Review form: Reviewer 1

Recommendation

Reject – article is scientifically unsound

Scientific importance: Is the manuscript an original and important contribution to its field?

Poor

General interest: Is the paper of sufficient general interest?

Good

Quality of the paper: Is the overall quality of the paper suitable?

Poor

Is the length of the paper justified?

Yes

Should the paper be seen by a specialist statistical reviewer?

No

Do you have any concerns about statistical analyses in this paper? If so, please specify them explicitly in your report.

Yes

It is a condition of publication that authors make their supporting data, code and materials available - either as supplementary material or hosted in an external repository. Please rate, if applicable, the supporting data on the following criteria.

Is it accessible?

Yes

Is it clear?

Yes

Is it adequate?

Yes

Do you have any ethical concerns with this paper?

No

Comments to the Author

Onstein and coauthors suggest that species richness of trichromatic primates coincides with richness of palms producing reddish fruits. They argue that this is a test of the longstanding hypotheses that trichromacy has evolved as an adaptation for frugivory. Their dataset of palm species is impressive and I think the authors have an interesting result regarding the abundance of palms and radiation of primate species but the current framing as a major test of the foraging hypothesis of primate trichromacy is not justified. Below I list a few of the reasons for my assessment:

The authors mention the application of an “unprecedented” dataset of > 400 primate species. However, all monkeys and apes in Africa and Asia are routine trichromats due to a single genetic duplication and divergence of the OPN1LW gene. Adding many catarrhine species and suggesting that these datapoints are independent is problematic and not appropriate (for example Hypothesis 1 and Hypothesis 2).

There is no discussion about the proportion of palms or even fruits included in the diets of the 400 primates included in the text of the manuscript.

Hypothesis 3 makes sweeping claims about fruit abundance in different regions, but the authors do not offer strong evidence to substantiate this.

No objective assessment of fruit conspicuity is used, rather classifications are subjective. I understand reflectance data might not be available for these palm species, but the subjective classifications are problematic. For example, palms with purple fruits are considered by the authors to be conspicuous to trichromats but these palms are likely chromatically conspicuous to dichromatic visual systems given the bluish component. Additionally, purple fruits are typically very dark, which would give a strong achromatic contrast against leaves that would be visible to trichromats, dichromats and monochromats.

Furthermore, aspects of the background are factually incorrect, which causes me to question how well the authors understand their study system. For example:

Line 64 - The authors make arguments about primate seed dispersal and cite gap width of frugivores as a constraint on fruit consumed – but they cite references to birds. Primates often consume fruit much larger than their mouths, which is facilitated by their teeth and extractive

strategies. In their next paragraph, they raise this point themselves - "Primates often disperse large fruits" but this also includes fruits that are relatively large relative to their gape width.

Line 84 - The authors state night active primates are monochromatic and do not include fruits in their diets. Many species of night active lemurs as well as owl monkeys eat substantial quantities of fruit and many species of night active primates are dichromatic, not monochromatic (e.g. mouse lemurs, aye-ayes).

Minor points:

The use of trichromats to only refer to day-active, frugivorous primates is unnecessary and introduces assumptions and differs from all other papers (hundreds) published on primate color vision.

Aspects of the structure and flow could be optimized. For example, discussion of fruit colors and primate color vision in line 81 is repetitive.

Review form: Reviewer 2 (Daniel Osorio)

Recommendation

Accept as is

Scientific importance: Is the manuscript an original and important contribution to its field?

Excellent

General interest: Is the paper of sufficient general interest?

Good

Quality of the paper: Is the overall quality of the paper suitable?

Excellent

Is the length of the paper justified?

Yes

Should the paper be seen by a specialist statistical reviewer?

Yes

Do you have any concerns about statistical analyses in this paper? If so, please specify them explicitly in your report.

Yes

It is a condition of publication that authors make their supporting data, code and materials available - either as supplementary material or hosted in an external repository. Please rate, if applicable, the supporting data on the following criteria.

Is it accessible?

N/A

Is it clear?

N/A

Is it adequate?

N/A

Do you have any ethical concerns with this paper?

No

Comments to the Author

There is longstanding interest in the co-evolutionary relationships between the colours of fruit and flowers and the colour vision of their animal dispersers. Primate trichromacy is an especially interesting case study, because it comes in two forms 'routine' and 'allelic', both of which have evolved independently at least twice and under positive selection. It is easy to show theoretically that trichromacy would be beneficial for finding the colourful fruit which are eaten and dispersed by many primates, and this prediction has good experimental support. There is also evidence that certain types of fruit have evolved to be dispersed by primates, which tend to be orange or yellow compared to bird or bat dispersed fruit. However the evidence that fruit colours are a major selective factor for primate trichromacy is weak and there are good grounds to question it. This study takes a macroevolutionary and macroecological approach, making the case at least in Africa for temporal and spatial link between the colourful palm fruit and the presence of primate dispersers. Although far from being the last word - and raising many questions, this is in my view the most original and substantive contribution to this field for many years. The work is of broad interest to those working on plant-animal interactions and the co-evolution sensory systems and biological communication signals.

Questions that immediately come to mind are whether species richness is a good measure of ecological impact, and how does the allelic trichromacy of most neotropical and at least some Malagasy primates inform the arguments here?

Review form: Reviewer 3

Recommendation

Major revision is needed (please make suggestions in comments)

Scientific importance: Is the manuscript an original and important contribution to its field?

Acceptable

General interest: Is the paper of sufficient general interest?

Acceptable

Quality of the paper: Is the overall quality of the paper suitable?

Poor

Is the length of the paper justified?

Yes

Should the paper be seen by a specialist statistical reviewer?

No

Do you have any concerns about statistical analyses in this paper? If so, please specify them explicitly in your report.

No

It is a condition of publication that authors make their supporting data, code and materials available - either as supplementary material or hosted in an external repository. Please rate, if applicable, the supporting data on the following criteria.

Is it accessible?

Yes

Is it clear?

Yes

Is it adequate?

Yes

Do you have any ethical concerns with this paper?

No

Comments to the Author

In this manuscript, Onstein and colleagues explore the relationships between the primate color vision frugivory on a global scale by comparing the trichromat primate species richness and the distributions of conspicuous fruits. They further use ancestral state reconstructions to look at the coevolution of the diversity of trichromat primate species and conspicuous palm fruit species. The methodologies the authors are using represent a novel way of approaching primate trichromacy that I haven't seen applied to the question before, and the large data set they have assembled is admirable.

Yet, while I think the methods and approach used are cool, I found the way the authors treated "trichromacy" and defined "trichromatic species" to be very problematic and not well explained. In the Introduction, they present a simplified view of primate trichromacy as primarily occurring in the Old World monkeys and apes (line 71), ignoring the widespread presence of polymorphic trichromacy in primates in the Americas and in several day-active species in Madagascar. Polymorphic trichromacy is where some females in a species are trichromats while all males and remaining females are dichromats (due to allelic variation at the LWS opsin gene). The proportion of individuals in a species that are trichromats can vary depending in part on the number of LWS opsin alleles that are present in the species.

While it is possible that different selective pressures shaped the evolution of polymorphic trichromacy compared to the "routine trichromacy" of the Old World species, the authors do not offer any explanation as to why they exclude polymorphic species as "trichromats" in their analyses. This is problematic, because many of the studies in the literature that have discussed the benefit of trichromacy for detecting conspicuous foods, like fruits, have been performed on New World polymorphic species. Two of these previous studies (e.g., Regan et al. 2001; Melin et al. 2017) are in fact cited by the authors of this manuscript in the Introduction.

Because of this oversight in how they define "trichromat species", I had trouble trusting their measures of "trichromat richness" and subsequent results. I do appreciate that they discuss polymorphic species in some parts of the Results, such as when they report that the proportion of conspicuous fruits was correlated with polymorphic species richness, although to a weaker extent. I wonder whether it could be beneficial to reframe the manuscript to emphasize that there are different types of primate trichromacy, and there is an effect of conspicuous fruits on both types of trichromacy (although weaker for polymorphic) – e.g., analyze both types separately and together for all of the methods? It just felt disingenuous in the Introduction to not mention polymorphic trichromacy at all, despite its widespread prevalence.

If the authors only want to test effects of conspicuous fruits on the species richness of routine trichromats, they should specify that and justify the decision. They should also acknowledge that routine trichromats are phylogenetically constrained to Africa and Asia (but are not found in Madagascar), except for the convergent evolution of routine trichromacy in one genus of New World monkey (*Alouatta*). I am unfamiliar with structural equation modeling – do you also need to take phylogenetic history into account when determining trichromat species richness?

Other concerns:

1. Primate species color vision classification needs to be updated. All Eulemur species except *E.*

flavifrons and *E. macaco* are dichromats (Jacobs et al. 2019).

2. One of the major hypotheses proposed to explain routine trichromacy is the exploitation of protein-rich young red leaves by catarrhine primates during periods of food scarcity (as a fallback resource). I felt that this hypothesis was not given sufficient attention. I was also curious as to whether the authors observed an increase in diurnal folivorous trichromat species from 10 Ma onward. They mention that there is an increase in diurnal frugivorous trichromats but not in non-trichromat species. The colobine subfamily of Cercopithecidae primates are all routine trichromats and have all evolved specialized stomachs for folivory. If there was an increase in both frugivorous and folivorous trichromats, it might be difficult to link the increase in species richness specifically with fruit coloration rather than an adaptive radiation of cercopithecids in general during the last 10 Ma.

Decision letter (RSPB-2019-2114.R0)

21-Oct-2019

Dear Dr Onstein:

I am writing to inform you that your manuscript RSPB-2019-2114 entitled "Fruit colours determine the broad-scale distribution and diversification of primate colour vision" has, in its current form, been rejected for publication in *Proceedings B*.

This action has been taken on the advice of referees, who have recommended that substantial revisions are necessary. With this in mind we would be happy to consider a resubmission, provided the comments of the referees are fully addressed. However please note that this is not a provisional acceptance.

Sincerely,

Dr Sasha Dall

Associate Editor

Comments to Author:

Three expert reviewers have now seen your paper, and while all of them acknowledge that your approach for adding weight to the well known-evolutionary relationship between primate trichromacy and fruit colour is both novel and admirable (with a remarkable data set), they express serious concerns (particularly reviewers 1 and 3) about several of your hypotheses and approaches (although Reviewers 2 and 3 do admit to being unfamiliar with structural equation modelling). Reviewer 1, for instance questions how appropriate it is to treat each primate species in Africa and Asia as independent data points, since all of these species are routine trichromats due to a single genetic duplication event in their evolutionary history. This reviewer also complains of conceptual and factual errors. Reviewer 3 has serious concerns with how you have apparently simplified the issue of primate trichromacy in framing your manuscript, and was also disappointed that you barely addressed the major competing hypothesis for routine trichromacy in Old World monkeys and apes, namely the exploitation of protein-rich young leaves as fallback foods. Thus the manuscript requires substantial revision and a careful and considered response to each of the comments and criticisms of the three referees.

Reviewer(s)' Comments to Author:

Referee: 1

Comments to the Author(s)

Onstein and coauthors suggest that species richness of trichromatic primates coincides with richness of palms producing reddish fruits. They argue that this is a test of the longstanding hypotheses that trichromacy has evolved as an adaptation for frugivory. Their dataset of palm species is impressive and I think the authors have an interesting result regarding the abundance of palms and radiation of primate species but the current framing as a major test of the foraging hypothesis of primate trichromacy is not justified. Below I list a few of the reasons for my assessment:

The authors mention the application of an “unprecedented” dataset of > 400 primate species. However, all monkeys and apes in Africa and Asia are routine trichromats due to a single genetic duplication and divergence of the OPN1LW gene. Adding many catarrhine species and suggesting that these datapoints are independent is problematic and not appropriate (for example Hypothesis 1 and Hypothesis 2).

There is no discussion about the proportion of palms or even fruits included in the diets of the 400 primates included in the text of the manuscript.

Hypothesis 3 makes sweeping claims about fruit abundance in different regions, but the authors do not offer strong evidence to substantiate this.

No objective assessment of fruit conspicuity is used, rather classifications are subjective. I understand reflectance data might not be available for these palm species, but the subjective classifications are problematic. For example, palms with purple fruits are considered by the authors to be conspicuous to trichromats but these palms are likely chromatically conspicuous to dichromatic visual systems given the bluish component. Additionally, purple fruits are typically very dark, which would give a strong achromatic contrast against leaves that would be visible to trichromats, dichromats and monochromats.

Furthermore, aspects of the background are factually incorrect, which causes me to question how well the authors understand their study system. For example:

Line 64 - The authors make arguments about primate seed dispersal and cite gap width of frugivores as a constraint on fruit consumed – but they cite references to birds. Primates often

consume fruit much larger than their mouths, which is facilitated by their teeth and extractive strategies. In their next paragraph, they raise this point themselves – “Primates often disperse large fruits” but this also includes fruits that are relatively large relative to their gape width.

Line 84 – The authors state night active primates are monochromatic and do not include fruits in their diets. Many species of night active lemurs as well as owl monkeys eat substantial quantities of fruit and many species of night active primates are dichromatic, not monochromatic (e.g. mouse lemurs, aye-ayes).

Minor points:

The use of trichromats to only refer to day-active, frugivorous primates is unnecessary and introduces assumptions and differs from all other papers (hundreds) published on primate color vision.

Aspects of the structure and flow could be optimized. For example, discussion of fruit colors and primate color vision in line 81 is repetitive.

Referee: 2

Comments to the Author(s)

There is longstanding interest in the co-evolutionary relationships between the colours of fruit and flowers and the colour vision of their animal dispersers. Primate trichromacy is an especially interesting case study, because it comes in two forms 'routine' and 'allelic', both of which have evolved independently at least twice and under positive selection. It is easy to show theoretically that trichromacy would be beneficial for finding the colourful fruit which are eaten and dispersed by many primates, and this prediction has good experimental support. There is also evidence that certain types of fruit have evolved to be dispersed by primates, which tend to be orange or yellow compared to bird or bat dispersed fruit. However the evidence that fruit colours are a major selective factor for primate trichromacy is weak and there are good grounds to question it. This study takes a macroevolutionary and macroecological approach, making the case at least in Africa for temporal and spatial link between the colourful palm fruit and the presence of primate dispersers. Although far from being the last word - and raising many questions, this is in my view the most original and substantive contribution to this field for many years. The work is of broad interest to those working on plant-animal interactions and the co-evolution sensory systems and biological communication signals.

Questions that immediately come to mind are whether species richness is a good measure of ecological impact, and how does the allelic trichromacy of most neotropical and at least some Malagasy primates inform the arguments here?

Referee: 3

Comments to the Author(s)

In this manuscript, Onstein and colleagues explore the relationships between the primate color vision frugivory on a global scale by comparing the trichromat primate species richness and the distributions of conspicuous fruits. They further use ancestral state reconstructions to look at the coevolution of the diversity of trichromat primate species and conspicuous palm fruit species. The methodologies the authors are using represent a novel way of approaching primate trichromacy that I haven't seen applied to the question before, and the large data set they have assembled is admirable.

Yet, while I think the methods and approach used are cool, I found the way the authors treated “trichromacy” and defined “trichromatic species” to be very problematic and not well explained. In the Introduction, they present a simplified view of primate trichromacy as primarily occurring

in the Old World monkeys and apes (line 71), ignoring the widespread presence of polymorphic trichromacy in primates in the Americas and in several day-active species in Madagascar. Polymorphic trichromacy is where some females in a species are trichromats while all males and remaining females are dichromats (due to allelic variation at the LWS opsin gene). The proportion of individuals in a species that are trichromats can vary depending in part on the number of LWS opsin alleles that are present in the species.

While it is possible that different selective pressures shaped the evolution of polymorphic trichromacy compared to the “routine trichromacy” of the Old World species, the authors do not offer any explanation as to why they exclude polymorphic species as “trichromats” in their analyses. This is problematic, because many of the studies in the literature that have discussed the benefit of trichromacy for detecting conspicuous foods, like fruits, have been performed on New World polymorphic species. Two of these previous studies (e.g., Regan et al. 2001; Melin et al. 2017) are in fact cited by the authors of this manuscript in the Introduction.

Because of this oversight in how they define “trichromat species”, I had trouble trusting their measures of “trichromat richness” and subsequent results. I do appreciate that they discuss polymorphic species in some parts of the Results, such as when they report that the proportion of conspicuous fruits was correlated with polymorphic species richness, although to a weaker extent. I wonder whether it could be beneficial to reframe the manuscript to emphasize that there are different types of primate trichromacy, and there is an effect of conspicuous fruits on both types of trichromacy (although weaker for polymorphic) – e.g., analyze both types separately and together for all of the methods? It just felt disingenuous in the Introduction to not mention polymorphic trichromacy at all, despite its widespread prevalence.

If the authors only want to test effects of conspicuous fruits on the species richness of routine trichromats, they should specify that and justify the decision. They should also acknowledge that routine trichromats are phylogenetically constrained to Africa and Asia (but are not found in Madagascar), except for the convergent evolution of routine trichromacy in one genus of New World monkey (*Alouatta*). I am unfamiliar with structural equation modeling – do you also need to take phylogenetic history into account when determining trichromat species richness?

Other concerns:

1. Primate species color vision classification needs to be updated. All *Eulemur* species except *E. flavifrons* and *E. macaco* are dichromats (Jacobs et al. 2019).
2. One of the major hypotheses proposed to explain routine trichromacy is the exploitation of protein-rich young red leaves by catarrhine primates during periods of food scarcity (as a fallback resource). I felt that this hypothesis was not given sufficient attention. I was also curious as to whether the authors observed an increase in diurnal folivorous trichromat species from 10 Ma onward. They mention that there is an increase in diurnal frugivorous trichromats but not in non-trichromat species. The colobine subfamily of Cercopithecidae primates are all routine trichromats and have all evolved specialized stomachs for folivory. If there was an increase in both frugivorous and folivorous trichromats, it might be difficult to link the increase in species richness specifically with fruit coloration rather than an adaptive radiation of cercopithecids in general during the last 10 Ma.

Author's Response to Decision Letter for (RSPB-2019-2114.R0)

See Appendix A.

RSPB-2019-2731.R0

Review form: Reviewer 1 (Amanda Melin)

Recommendation

Major revision is needed (please make suggestions in comments)

Scientific importance: Is the manuscript an original and important contribution to its field?

Excellent

General interest: Is the paper of sufficient general interest?

Excellent

Quality of the paper: Is the overall quality of the paper suitable?

Good

Is the length of the paper justified?

No

Should the paper be seen by a specialist statistical reviewer?

No

Do you have any concerns about statistical analyses in this paper? If so, please specify them explicitly in your report.

No

It is a condition of publication that authors make their supporting data, code and materials available - either as supplementary material or hosted in an external repository. Please rate, if applicable, the supporting data on the following criteria.

Is it accessible?

Yes

Is it clear?

No

Is it adequate?

No

Do you have any ethical concerns with this paper?

No

Comments to the Author

The manuscript is much improved and that the new analyses contribute significant findings to the previous research. This is an exciting dataset and I feel the authors have uncovered several interesting results. However, I still feel that not all of the conclusions (including a main conclusion) are justified by the results, and I think the authors can and should highlight discoveries they have made that they are not currently focusing on.

The first conclusion in the last sentence of the abstract is supported by the results (co-diversification of palms and trichromats) but the second claim is not justified by the results. Given that genetic evidence shows a single origin of trichromacy in catarrhines (the diurnal primates currently on mainland Africa) co-diversification of palms across Africa happened once

routine trichromacy was established – Aspects shaping the radiation of primate species after this event cannot explain the evolution of trichromacy.

The authors do have evidence that trichromats might have selected for radiation of colorful (vs cryptic) palms – and this would be a very interesting and exciting result that should be given more spotlight. That similar patterns are not found in polymorphic species suggests to be those primates don't shape fruit traits in the same way – i.e. the evolutionary relationships between primates and palms are impacted by primate color vision.

Overall - the original selective pressure for trichromacy is not clarified by the present analysis. If the authors could show conspicuous palms were more common in areas where first catarrhines evolved but not where the polymorphic trichromats evolved this would be more direct evidence. I'm not suggesting this is possible, just giving an example that I hope clarifies my point.

Overall - I feel the paper needs reworking throughout to clarify what is being tested and what can be concluded. But to be clear – I think the analyses and results are comprehensive, novel and exciting – I feel they are just making a slightly different contribution that the present focus of the manuscript.

Minor points –

Use “palms” or “palm fruits” and not just “fruits” throughout. It is misleading to equate “palm” with “fruit” – this is done throughout the MS and should be changed, or it will lead to misquoting. (I agree palms are important, but they are a small fraction of total fruits).

Line 67 – “frugivory hypothesis” and not “foraging” hypothesis. Leaf eating is also “foraging” and a distinct hypothesis.

Line 103 – 115 The three “hypotheses” given by the authors are all predictions stemming from a single hypothesis that primate color vision and fruit traits are related.

Please organize the results and discussion into a structure that follows the predictions (at present 3 predictions/ hypotheses but there are 4 sections (a-d) in the results and 2 sections (a-b) in the discussion. This disrupts the flow of the paper and makes it hard to follow.

Line 165 - How are cathemeral and crepuscular primate species handled in your diurnal vs nocturnal scenario? This is an important consideration, especially considering the variation between and between species and regions in Madagascar (some lemurs are nocturnal at some sights, and cathemeral at others, for example). The methods for assigning primate activity patterns should be acknowledged clearly and stated how the authors assigned categories.

The discussion is overlong and could be usefully shortened.

I'd be happy to look at another version of the manuscript.

Best wishes,
Amanda Melin

Review form: Reviewer 3

Recommendation

Accept with minor revision (please list in comments)

Scientific importance: Is the manuscript an original and important contribution to its field?
Excellent

General interest: Is the paper of sufficient general interest?
Good

Quality of the paper: Is the overall quality of the paper suitable?
Good

Is the length of the paper justified?
Yes

Should the paper be seen by a specialist statistical reviewer?
No

Do you have any concerns about statistical analyses in this paper? If so, please specify them explicitly in your report.
No

It is a condition of publication that authors make their supporting data, code and materials available - either as supplementary material or hosted in an external repository. Please rate, if applicable, the supporting data on the following criteria.

Is it accessible?
Yes

Is it clear?
Yes

Is it adequate?
Yes

Do you have any ethical concerns with this paper?
No

Comments to the Author

In this manuscript, Onstein and colleagues use novel methods to explore a longstanding question in mammal vision – what factors have influenced the evolution of trichromatic color vision in primate clades? I reviewed a previous version of this manuscript, and it is very clear that the authors took the reviewer comments to heart and made substantive changes to the manuscript that have greatly improved it. I think this manuscript now offers an exciting and novel contribution to the research area and employs really cool methodology.

I have only minor suggestions for revisions. Primarily, these involve citing statements more thoroughly, clarifying some sentences, and fixing a typo.

- Lines 77-80: you should include citations or at least a citation for the types/distribution of color vision. Gerald Jacobs has great reviews you could cite.
- Lines 213-215: how did you estimate the dates for the evolution of polymorphic trichromacy and routine trichromacy? I think you should include the citation for the phylogeny you use in the main text and not just supplemental.
- Lines 251-254: “Although we cannot exclude this possibility, and detailed spatial and phylogenetic data on leaf coloration may allow the testing of these competing hypotheses in the future, results from our sensitivity analyses ...” This sentence is fairly convoluted and it’s hard to follow. It may work better if you break it up or expand it. Similarly, I think you should expand what you mean regarding the new sentence from 267-269.

- Line 264: lowercase "A" for Although, since the sentence begins "Indeed, although ..."

Decision letter (RSPB-2019-2731.R0)

13-Dec-2019

Dear Dr Onstein:

Your manuscript has now been peer reviewed and the reviews have been assessed by an Associate Editor. The reviewers' comments (not including confidential comments to the Editor) and the comments from the Associate Editor are included at the end of this email for your reference. As you will see, the reviewers and the Editors have raised some concerns with your manuscript and we would like to invite you to revise your manuscript to address them.

Research ethics:

Use of animals and field studies:

It is a condition of publication that you make available the data and research materials supporting the results in the article. Datasets should be deposited in an appropriate publicly

available repository and details of the associated accession number, link or DOI to the datasets must be included in the Data Accessibility section of the article (<https://royalsociety.org/journals/ethics-policies/data-sharing-mining/>). Reference(s) to datasets should also be included in the reference list of the article with DOIs (where available).

Please submit a copy of your revised paper within three weeks. If we do not hear from you within this time your manuscript will be rejected. If you are unable to meet this deadline please let us know as soon as possible, as we may be able to grant a short extension.

Best wishes,
Dr Sasha Dall
mailto: proceedingsb@royalsociety.org

Associate Editor Board Member

Comments to Author:

Two referees have now seen the revised version of your manuscript and as you will see both of them were very pleased with the efforts you made to address their previous comments and criticisms. Moreover, both of them feel that your contribution is important and interesting. Nonetheless both have a number of suggestions for further improvement. In particular, Reviewer 1 still has issues with some of your conclusions not being supported by the data (including one of the main conclusions) and also feels that you under-sell yourselves when presenting other data that have very interesting implications for the evolution of colour vision. I thus recommend that you carefully consider these comments and suggestions and adjust your text accordingly.

Reviewer(s)' Comments to Author:

Referee: 1

Comments to the Author(s).

The manuscript is much improved and that the new analyses contribute significant findings to the previous research. This is an exciting dataset and I feel the authors have uncovered several interesting results. However, I still feel that not all of the conclusions (including a main conclusion) are justified by the results, and I think the authors can and should highlight discoveries they have made that they are not currently focusing on.

The first conclusion in the last sentence of the abstract is supported by the results (co-diversification of palms and trichromats) but the second claim is not justified by the results. Given that genetic evidence shows a single origin of trichromacy in catarrhines (the diurnal primates currently on mainland Africa) co-diversification of palms across Africa happened once routine trichromacy was established – Aspects shaping the radiation of primate species after this event cannot explain the evolution of trichromacy.

The authors do have evidence that trichromats might have selected for radiation of colorful (vs cryptic) palms – and this would be a very interesting and exciting result that should be given more spotlight. That similar patterns are not found in polymorphic species suggests to be those primates don't shape fruit traits in the same way – i.e. the evolutionary relationships between primates and palms are impacted by primate color vision.

Overall - the original selective pressure for trichromacy is not clarified by the present analysis. If the authors could show conspicuous palms were more common in areas where first catarrhines evolved but not where the polymorphic trichromats evolved this would be more direct evidence. I'm not suggesting this is possible, just giving an example that I hope clarifies my point.

Overall - I feel the paper needs reworking throughout to clarify what is being tested and what can be concluded. But to be clear – I think the analyses and results are comprehensive, novel and exciting – I feel they are just making a slightly different contribution that the present focus of the manuscript.

Minor points –

Use “palms” or “palm fruits” and not just “fruits” throughout. It is misleading to equate “palm” with “fruit” – this is done throughout the MS and should be changed, or it will lead to misquoting. (I agree palms are important, but they are a small fraction of total fruits).

Line 67 – “frugivory hypothesis” and not “foraging” hypothesis. Leaf eating is also “foraging” and a distinct hypothesis.

Line 103 – 115 The three “hypotheses” given by the authors are all predictions stemming from a single hypothesis that primate color vision and fruit traits are related.

Please organize the results and discussion into a structure that follows the predictions (at present 3 predictions/ hypotheses but there are 4 sections (a-d) in the results and 2 sections (a-b) in the discussion. This disrupts the flow of the paper and makes it hard to follow.

Line 165 - How are cathemeral and crepuscular primate species handled in your diurnal vs nocturnal scenario? This is an important consideration, especially considering the variation between and between species and regions in Madagascar (some lemurs are nocturnal at some sights, and cathemeral at others, for example). The methods for assigning primate activity patterns should be acknowledged clearly and stated how the authors assigned categories.

The discussion is overlong and could be usefully shortened.

I'd be happy to look at another version of the manuscript.

Best wishes,
Amanda Melin

Referee: 3

Comments to the Author(s).

In this manuscript, Onstein and colleagues use novel methods to explore a longstanding question in mammal vision – what factors have influenced the evolution of trichromatic color vision in primate clades? I reviewed a previous version of this manuscript, and it is very clear that the authors took the reviewer comments to heart and made substantive changes to the manuscript that have greatly improved it. I think this manuscript now offers an exciting and novel contribution to the research area and employs really cool methodology.

I have only minor suggestions for revisions. Primarily, these involve citing statements more thoroughly, clarifying some sentences, and fixing a typo.

- Lines 77-80: you should include citations or at least a citation for the types/distribution of color vision. Gerald Jacobs has great reviews you could cite.
- Lines 213-215: how did you estimate the dates for the evolution of polymorphic trichromacy and routine trichromacy? I think you should include the citation for the phylogeny you use in the main text and not just supplemental.
- Lines 251-254: “Although we cannot exclude this possibility, and detailed spatial and phylogenetic data on leaf coloration may allow the testing of these competing hypotheses in the future, results from our sensitivity analyses ...” This sentence is fairly convoluted and it’s hard to follow. It may work better if you break it up or expand it. Similarly, I think you should expand what you mean regarding the new sentence from 267-269.
- Line 264: lowercase “A” for Although, since the sentence begins “Indeed, although ...”

Author's Response to Decision Letter for (RSPB-2019-2731.R0)

See Appendix B.

RSPB-2019-2731.R1 (Revision)

Review form: Reviewer 1 (Amanda Melin)

Recommendation

Accept with minor revision (please list in comments)

Scientific importance: Is the manuscript an original and important contribution to its field?

Excellent

General interest: Is the paper of sufficient general interest?

Excellent

Quality of the paper: Is the overall quality of the paper suitable?

Good

Is the length of the paper justified?

Yes

Should the paper be seen by a specialist statistical reviewer?

No

Do you have any concerns about statistical analyses in this paper? If so, please specify them explicitly in your report.

No

It is a condition of publication that authors make their supporting data, code and materials available - either as supplementary material or hosted in an external repository. Please rate, if applicable, the supporting data on the following criteria.

Is it accessible?

Yes

Is it clear?

Yes

Is it adequate?

Yes

Do you have any ethical concerns with this paper?

No

Comments to the Author

The current title maintains adherence to the previous "causation" framing, which the authors agreed was flawed, and should be revised to something like: Palm fruit colors are correlated with the broad-scale distribution and diversification of primate color vision phenotypes.

Minor suggestions are included as tracked changes in the word document.

Supplementary Methods (No line numbers were provided, so hopefully these edits are easy to locate)

Page 1

Given some of these data are interpolated from higher taxonomic levels, revise the opening sentences to: We collected or inferred/ interpolated functional trait data from.....

I suggest you remove the following as you don't focus on the tuning of pigments: "Although the precision of the collected vision data was much higher at the species level compared to the family level or the interpolated level, distinct primate lineages have a conservative spectral tuning of their photopigments (SurrIDGE et al., 2003). "

Page 1-2

Of the fruit species you classify as "cryptic" not only the purple, but also white and black and blue and essentially all non-green colors would likely be discriminable by trichromats and

dichromats due to luminance difference. "Cryptic" (suggestive of color-matching with leaves) is therefore not the best name. I suggest you call them "Other" or "Non-conspicuous" in which this category includes greenish cryptic fruits. So, you would have 2 categories: "Conspicuous to trichromats (red, orange, yellow, conspicuous for short)" and "Other" (Green, blue, purple, black, white, ivory etc etc). Please revised accordingly here and in main text and Appendix S2.

Suggest avoiding colonial terms of "New World/ Neotropical" and "Old World/ Paleotropical" and describe them as monkeys of the Americas or African and Asian Primates etc.

The revisions to remove the previous framing around testing the frugivory hypothesis were not entirely thorough, leaving the flow of the introduction a bit disjointed. The authors should at minimum add in their introduction a few sentences in opening paragraph(s) concerning the potential for dispersers to shape fruit communities and remove continued adherence to old framing – a few listed below:

Title (see above): Suggestion: Palm fruit colors are correlated with the broad-scale distribution and diversification of primate color vision phenotypes.

Line 312 – are consistent with, is more appropriate than "strongly support"

End of Supplementary Methods: "In conclusion, the results including figs indicate that figs, as compared to palms, probably played a less important role for trichromat distribution and diversification in mainland Africa (also see Dominy et al., 2003)." (Please revise)

The opening sentence of abstract – you don't test the hypothesis that you start with in your abstract. I suggest you are consistent with the hypothesis you adopt in the text of your revised paper and quote in the response to reviewers: "that the origin, distribution and diversity of trichromatic vision in primates is positively associated with the availability of conspicuous fruits of palms (Arecaceae)"

Decision letter (RSPB-2019-2731.R1)

31-Jan-2020

Dear Dr Onstein

I am pleased to inform you that your Review manuscript RSPB-2019-2731.R1 entitled "Palm fruit colours determine the broad-scale distribution and diversification of primate colour vision" has been accepted for publication in Proceedings B.

The referee(s) do not recommend minor changes. Because the schedule for publication is very tight, it is a condition of publication that you submit the revised version of your manuscript within 7 days. If you do not think you will be able to meet this date please let me know immediately.

To upload your manuscript, log into <http://mc.manuscriptcentral.com/prsb> and enter your Author Centre, where you will find your manuscript title listed under "Manuscripts with Decisions." Under "Actions," click on "Create a Revision." Your manuscript number has been appended to denote a revision.

You will be unable to make your revisions on the originally submitted version of the manuscript. Instead, upload a new version through your Author Centre.

1) A text file of the manuscript (doc, txt, rtf or tex), including the references, tables (including captions) and figure captions. Please remove any tracked changes from the text before submission. PDF files are not an accepted format for the "Main Document".

2) A separate electronic file of each figure (tiff, EPS or print-quality PDF preferred). The format should be produced directly from original creation package, or original software format. Please note that PowerPoint files are not accepted.

3) Electronic supplementary material: this should be contained in a separate file from the main text and the file name should contain the author's name and journal name, e.g. authorname_procb_ESM_figures.pdf

All supplementary materials accompanying an accepted article will be treated as in their final form. They will be published alongside the paper on the journal website and posted on the online figshare repository. Files on figshare will be made available approximately one week before the accompanying article so that the supplementary material can be attributed a unique DOI. Please see: <https://royalsociety.org/journals/authors/author-guidelines/>

4) Data-Sharing and data citation

It is a condition of publication that data supporting your paper are made available. Data should be made available either in the electronic supplementary material or through an appropriate repository. Details of how to access data should be included in your paper. Please see <https://royalsociety.org/journals/ethics-policies/data-sharing-mining/> for more details.

<http://datadryad.org/submit?journalID=RSPB&manu=RSPB-2019-2731.R1> which will take you to your unique entry in the Dryad repository.

Once again, thank you for submitting your manuscript to Proceedings B and I look forward to receiving your final version. If you have any questions at all, please do not hesitate to get in touch.

Sincerely,

Dr Sasha Dall

Associate Editor

Comments to Author:

I am pleased to inform you that the reviewer is satisfied that you have now addressed their concerns and feels that your manuscript is almost ready for publication. The reviewer recommends however that you make some minor revisions and adjust the text prior to final re-submission, reporting "that the changes to the framing were rather incremental, such that the logic and flow of the introduction is a bit halting". If you can make these final minor revisions (some of which were included by the reviewer in your track-changed manuscript), then my feeling is that your manuscript can be finally published!

Reviewer(s)' Comments to Author:

Referee: 1

Comments to the Author(s)

The current title maintains adherence to the previous "causation" framing, which the authors agreed was flawed, and should be revised to something like: Palm fruit colors are correlated with the broad-scale distribution and diversification of primate color vision phenotypes.

Minor suggestions are included as tracked changes in the word document.

Supplementary Methods (No line numbers were provided, so hopefully these edits are easy to locate)

Page 1

Given some of these data are interpolated from higher taxonomic levels, revise the opening sentences to: We collected or inferred/ interpolated functional trait data from.....

I suggest you remove the following as you don't focus on the tuning of pigments: "Although the precision of the collected vision data was much higher at the species level compared to the family level or the interpolated level, distinct primate lineages have a conservative spectral tuning of their photopigments (SurrIDGE et al., 2003)."

Page 1-2

Of the fruit species you classify as "cryptic" not only the purple, but also white and black and blue and essentially all non-green colors would likely be discriminable by trichromats and dichromats due to luminance difference. "Cryptic" (suggestive of color-matching with leaves) is therefore not the best name. I suggest you call them "Other" or "Non-conspicuous" in which this category includes greenish cryptic fruits. So, you would have 2 categories: "Conspicuous to trichromats (red, orange, yellow, conspicuous for short)" and "Other" (Green, blue, purple, black, white, ivory etc etc). Please revised accordingly here and in main text and Appendix S2.

Suggest avoiding colonial terms of "New World/ Neotropical" and "Old World/ Paleotropical" and describe them as monkeys of the Americas or African and Asian Primates etc.

The revisions to remove the previous framing around testing the frugivory hypothesis were not entirely thorough, leaving the flow of the introduction a bit disjointed. The authors should at minimum add in their introduction a few sentences in opening paragraph(s) concerning the potential for dispersers to shape fruit communities and remove continued adherence to old framing - a few listed below:

Title (see above): Suggestion: Palm fruit colors are correlated with the broad-scale distribution and diversification of primate color vision phenotypes.

Line 312 - are consistent with, is more appropriate than "strongly support"

End of Supplementary Methods: "In conclusion, the results including figs indicate that figs, as compared to palms, probably played a less important role for trichromat distribution and diversification in mainland Africa (also see Dominy et al., 2003)." (Please revise)

The opening sentence of abstract - you don't test the hypothesis that you start with in your abstract. I suggest you are consistent with the hypothesis you adopt in the text of your revised paper and quote in the response to reviewers: "that the origin, distribution and diversity of

trichromatic vision in primates is positively associated with the availability of conspicuous fruits of palms (Arecaceae))”

Author's Response to Decision Letter for (RSPB-2019-2731.R1)

See Appendix C.

Decision letter (RSPB-2019-2731.R2)

03-Feb-2020

Dear Dr Onstein

I am pleased to inform you that your manuscript entitled "Palm fruit colours are linked with the broad-scale distribution and diversification of primate colour vision systems" has been accepted for publication in Proceedings B.

Open Access

Paper charges

You are allowed to post any version of your manuscript on a personal website, repository or preprint server. However, the work remains under media embargo and you should not discuss it

with the press until the date of publication. Please visit <https://royalsociety.org/journals/ethics-policies/media-embargo> for more information.

Sincerely,
Proceedings B
<mailto:proceedingsb@royalsociety.org>

Appendix A

Associate Editor

Comments to Author:

Three expert reviewers have now seen your paper, and while all of them acknowledge that your approach for adding weight to the well known-evolutionary relationship between primate trichromacy and fruit colour is both novel and admirable (with a remarkable data set), they express serious concerns (particularly reviewers 1 and 3) about several of your hypotheses and approaches (although Reviewers 2 and 3 do admit to being unfamiliar with structural equation modelling). Reviewer 1, for instance questions how appropriate it is to treat each primate species in Africa and Asia as independent data points, since all of these species are routine trichromats due to a single genetic duplication event in their evolutionary history. This reviewer also complains of conceptual and factual errors. Reviewer 3 has serious concerns with how you have apparently simplified the issue of primate trichromacy in framing your manuscript, and was also disappointed that you barely addressed the major competing hypothesis for routine trichromacy in Old World monkeys and apes, namely the exploitation of protein-rich young leaves as fallback foods. Thus the manuscript requires substantial revision and a careful and considered response to each of the comments and criticisms of the three referees.

We thank the three referees for their positive feedback and the constructive comments. All suggestions were fully taken into account in the revision. In our response to the referees, we reply to all comments in detail (in blue) and indicate where and how we have dealt with them (the changed text parts are indicated in blue in the manuscript). Specifically, we implemented simulations to address issues with the single origin of trichromacy in Old World monkeys and apes (reviewer #1) and present the manuscript for polymorphic and routine trichromatic primates separately (and combined) (reviewer #3). We also expanded the discussion comparing our results with the competing hypothesis of young leaves (reviewer #3). Generally, results remained the same, but in addition we detected differences between polymorphic and routine trichromatic primates in relation to the distribution of conspicuous fruits which, we think, make the manuscript more interesting.

Reviewer(s)' Comments to Author:

Referee: 1

Comments to the Author(s)

Onstein and coauthors suggest that species richness of trichromatic primates coincides with richness of palms producing reddish fruits. They argue that this is a test of the longstanding hypotheses that trichromacy has evolved as an adaptation for frugivory. Their dataset of palm species is impressive and I think the authors have an interesting result regarding the abundance of palms and radiation of primate species but the current framing as a major test of the foraging hypothesis of primate trichromacy is not justified. Below I list a few of the reasons for my assessment:

We thank the reviewer for a very thorough review that raised very good and important points, and we hope that we have now dealt with these concerns in a satisfactory way.

The authors mention the application of an “unprecedented” dataset of > 400 primate species. However, all monkeys and apes in Africa and Asia are routine trichromats due to a single genetic duplication and divergence of the OPN1LW gene. Adding many catarrhine species and suggesting that these datapoints are independent is problematic and not appropriate (for example Hypothesis 1 and Hypothesis 2).

The reviewer raises an important point. It is true that the evolution of trichromatic vision only happened twice independently in primates, and to truly test an ‘adaptive’ hypothesis, we would need many independent origins to be able to consistently show that the evolution was adaptive to e.g. conspicuous fruits. We thus fully agree that with this study, we cannot prove that trichromatic vision is adaptive as such. However, we can test whether the global distribution of this trait coincides with the distribution of conspicuous fruits (hypothesis 1 and 2). Although not a direct test of adaptivity, a pattern that coincides does suggest that there may be an ecological relationship between the traits, especially if this pattern deviates from a random distribution. We therefore added a simulation analysis in which we simulated primate distributions 1000 times and repeated the SEMs, and then compared the observed effect of the proportion of conspicuous fruits on trichromatic and/or polymorphic species richness to the distribution of 1000 simulated effects. We show that the observed effect lies outside the 95% confidence interval of simulated effects and therefore strongly deviates from a random expectation (at least globally for routine trichromatic primates and in Africa). This means that places that are species-rich in trichromatic primates also show the highest proportions of conspicuous fruits. We further discuss this issue in the discussion, e.g.

L. 236 “Our results show that at a global scale, the geographical distribution of trichromatic primates is associated with the availability of conspicuous relative to cryptic palm fruits (Figure 2). However, when including polymorphic primates, our results do not deviate from a neutral expectation in which primates are randomly distributed across regions. This may be explained by the fact that colour vision systems evolved only few times independently within primates (probably < 7 times, Figure S11), and colour vision is thus geographically (Figure 2) and phylogenetically (Figure S11) constrained. Thus, our results strongly support the fruit foraging hypothesis (Allen, 1879) for routine trichromacy, especially in Africa (Figure 2, Figs. S7, S8), but the ecological determinants of polymorphic vision remain questionable.”

There is no discussion about the proportion of palms or even fruits included in the diets of the 400 primates included in the text of the manuscript.

Unfortunately, direct observational data on diet for all 400 primates is currently not available. This is why we took a macroecological approach that explores the co-occurrence of palms and primates at large spatial scales, and thus evaluates whether there is generality in the occurrence patterns between trichromats and conspicuous fruits. However, we have also explored the literature for direct diet data for primates, and found that there are several well-studied primate species across different biogeographical realms that are specifically known to include palm fruits in their diets (see references below and in the main text). We now included this in the introduction, and further explain the examples in the discussion (see main text, “(b) The macroecological drivers of routine trichromatic primate richness”):

L98: “Palms are keystone resources for frugivores and nearly all species are animal dispersed (mainly by birds, primates, bats and other mammals) (Zona & Henderson, 1989). Furthermore, there are numerous examples of palm-primate seed dispersal interactions (Homewood, 1978; Palacios et al., 1997; Tutin et al., 1997; Bentley-Condit, 2009; Schreier, 2010; Kulp & Heymann, 2015; Santhosh et al., 2015).”

Hypothesis 3 makes sweeping claims about fruit abundance in different regions, but the authors do not offer strong evidence to substantiate this.

Hypothesis 3 (now hypothesis 2) evaluates whether the relationship between conspicuous fruits and trichromatic species richness on the African mainland changes depending on which regions are included in the model. This is one way to evaluate whether the strength of the effect differs across climatic gradients, for example. Our model shows that the effect indeed changes depending on whether dry/subtropical regions are included in the model or not, which we think is interesting. It proposes that this relationship is strong in dry/subtropical

regions, and we further discuss this result in the discussion (L298, L257). We have tried to provide support from observational data why this effect may be stronger in these regions, linked to competition and the advantage of this colour vision system under these conditions (L298). We do not claim that palm fruits are more abundant in these regions, only that the proportion of conspicuous fruits (relative to cryptic fruit species) has a stronger effect on total trichromatic richness in these regions, even though total richness (and probably abundance) is higher in tropical rainforests, for example. We added a sentence to the discussion to clarify this.

L290: “Interestingly, even though total species richness of both palms and primates is highest in tropical regions, we found the strongest effect of conspicuous palm fruits on trichromatic richness in transition zones between arid and subtropical (African) forests (Figure 3, Figure S10).”

No objective assessment of fruit conspicuity is used, rather classifications are subjective. I understand reflectance data might not be available for these palm species, but the subjective classifications are problematic. For example, palms with purple fruits are considered by the authors to be conspicuous to trichromats but these palms are likely chromatically conspicuous to dichromatic visual systems given the bluish component. Additionally, purple fruits are typically very dark, which would give a strong achromatic contrast against leaves that would be visible to trichromats, dichromats and monochromats.

The reviewer is correct that no objective assessment for fruit colour was made, although we did explore whether the subjective classification of colours was consistent with reflectance measurements from 18 palm species (see fruit colour experiment in the SI). However, following the reviewer’s concerns, we have now removed purple palm fruits from the ‘conspicuous’ category (only very few palm species have purple fruits), and added them to the ‘cryptic’ category. We explain the reasoning for this in the supplementary methods (i.e. purple fruits would be equally visible to dichromats as to trichromats). We have run all analyses again and the results remained almost identical to the previous results. We present these new results in the main text now.

Furthermore, aspects of the background are factually incorrect, which causes me to question how well the authors understand their study system. For example:

Line 64 - The authors make arguments about primate seed dispersal and cite gape width of frugivores as a constraint on fruit consumed – but they cite references to birds. Primates often consume fruit much larger than their mouths, which is facilitated by their teeth and extractive strategies. In their next paragraph, they raise this point themselves – “Primates often disperse large fruits” but this also includes fruits that are relatively large relative to their gape width.

Thank you for pointing this out. Several of the authors have a background in palm ecology/diversity and one of the coauthors is a primatologist. We acknowledge that we are of course not specialists on all 400 primate species, but the general aim of the study was to investigate the ecological relationship between colour vision and fruit colour, which of course requires some basic knowledge about the species, but more importantly knowledge on the ecology of the seed dispersal system (which several of the authors have). We corrected the statements pointed out by the reviewer, see below.

The (previously) first paragraph gave a general introduction about how traits of fruits and frugivores influence the interaction between plants/ animals and seed dispersal services, in which we indeed cited the well-studied relationship between body size/gape width and fruit size, including references to birds and mammals more generally. Only in the second paragraph we focused on primates. Now, we restructured this first paragraph to avoid confusion and focused it on primates, thus removed the references to birds, for example.

As suggested by the reviewer, we now added the observation that fruits eaten by primates are often larger than expected from primate gape widths, due to their teeth and their ability to handle fruits with their hands:

L63: “Primates are important seed dispersers in tropical forests, particularly of large fruits (often larger than expected from primate gape widths), and fruit eating is facilitated by their teeth and hands (Chapman & Russo, 2007; Fleming & Kress, 2013).”

Line 84 – The authors state night active primates are monochromatic and do not include fruits in their diets. Many species of night active lemurs as well as owl monkeys eat substantial quantities of fruit and many species of night active primates are dichromatic, not monochromatic (e.g. mouse lemurs, aye-ayes).

We indicated that *many* night-active primates, such as the New World night monkeys and Asian lorises, are ‘monochromatic’ (black-and-white vision) and *often* do not include fruits in their diet – but based on the reviewers comment we removed this sentence now, as we do not provide the data to fully support this statement.

Minor points:

The use of trichromats to only refer to day-active, frugivorous primates is unnecessary and introduces assumptions and differs from all other papers (hundreds) published on primate color vision.

This is an interesting point, and we now added analyses including all primates, not only the day-active frugivorous ones. Initially, our rationale to focus on day-active, frugivorous trichromats was that to understand the ecological role of trichromatic colour vision in the dispersal of fruits with particular colours, it’s important to focus on those trichromats that actually interact with these fruits. As we do not have detailed observations on interactions for all 400 primate species, we limited the data to those primates that are known to be frugivorous and day-active to test our hypothesis, because those that aren’t, are unlikely to interact with the palms. It is an interesting comparison though to focus on all primates and compare the results to those that are day-active and frugivorous. In addition, as pointed out by reviewer #3, polymorphic species may need to be included as well because of their potential to also interact with the palm fruits that have reddish colours. Therefore, we investigate the hypotheses now for trichromatic primates, polymorphic primates and trichromatic and polymorphic primates combined. Interestingly, although results remain very similar when including polymorphic primates as well, these results are not robust when we compare the observed pattern to results from simulations in which we randomly reshuffle primate species across botanical countries, and repeat the analyses. This means that polymorphic vision (which preceded routine trichromacy in an evolutionary sense) may have evolved in response to different selection pressures than fruit colours, or at least several selection pressures may have been at play to maintain the variation of colour vision systems in these species, and determine the present-day distribution of these species (also see discussion).

Aspects of the structure and flow could be optimized. For example, discussion of fruit colors and primate color vision in line 81 is repetitive.

Thanks for pointing this out, we have now restructured the first two paragraphs to avoid repetition.

Referee: 2

Comments to the Author(s)

There is longstanding interest in the co-evolutionary relationships between the colours of fruit and flowers and the colour vision of their animal dispersers. Primate trichromacy is an especially interesting case study, because it comes in two forms 'routine' and 'allelic', both of which have evolved independently at least twice and under positive selection. It is easy to show theoretically that trichromacy would be beneficial for finding the colourful fruit which are eaten and dispersed by many primates, and this prediction has good experimental support. There is also evidence that certain types of fruit have evolved to be dispersed by primates, which tend to be orange or yellow compared to bird or bat dispersed fruit. However the evidence that fruit colours are a major selective factor for primate trichromacy is weak and there are good grounds to question it. This study takes a macroevolutionary and macroecological approach, making the case at least in Africa for temporal and spatial link between the colourful palm fruit and the presence of primate dispersers. Although far from being the last word - and raising many questions, this is in my view the most original and substantive contribution to this field for many years. The work is of broad interest to those working on plant-animal interactions and the co-evolution sensory systems and biological communication signals.

Thank you very much for these kind words!

Questions that immediately come to mind are whether species richness is a good measure of ecological impact, and how does the allelic trichromacy of most neotropical and at least some Malagasy primates inform the arguments here?

The reviewer indicates that species richness may not reflect ecological impact. That is an important point – we do not know whether higher primate species richness is correlated to higher primate abundance, and therefore increased seed dispersal services. We have added this important assumption to the conclusion now:

L328: “Our study provides novel evidence that the geographic distribution of trichromatic colour vision in Africa is associated with the broad-scale distribution of conspicuous fruits of palms. The macroecological and macroevolutionary analyses performed in this study do not prove causality, and additional evidence is needed to show that these co-occurrence patterns of palm fruit colours and primate species richness are related to actual seed dispersal interactions and therefore reflect ecological impact.”

The reviewer also raises questions about the difference between polymorphic vision and trichromatic vision, something that also came up in comments from reviewer #3. Polymorphic vision evolutionary preceded the evolution of trichromatic vision. In fact, it seems to more or less coincide with the evolution of conspicuous palm fruits (see Fig. S11). We therefore explored the results for polymorphic primates as well, to evaluate whether conspicuous fruits remain an important determinant of polymorphic primates globally. Interestingly, the effect of conspicuous fruits is weaker when polymorphic primates are included in the analysis (Fig. 2 and Fig. S2), and the effect does not deviate from a random expectation based on the simulation study we now have performed (results are shown in Fig. S3). This means that the distribution of polymorphic primates is not significantly linked to the availability of conspicuous fruits, and other (or additional) selective pressures may have influenced their distribution. This may also explain why part of the population has remained dichromatic over evolutionary time, suggesting that there may be balancing selection for both traits (trichromacy and dichromacy) to remain in the species. Similarly, the rapid Malagasy primate radiation seems to be independent from colour vision / fruit colour relationships (Figure S12).

Referee: 3

Comments to the Author(s)

In this manuscript, Onstein and colleagues explore the relationships between the primate color vision frugivory on a global scale by comparing the trichromat primate species richness and the distributions of conspicuous fruits. They further use ancestral state reconstructions to look at the coevolution of the diversity of trichromat primate species and conspicuous palm fruit species. The methodologies the authors are using represent a novel way of approaching primate trichromacy that I haven't seen applied to the question before, and the large data set they have assembled is admirable.

Thank you very much for your kind words and thorough review.

Yet, while I think the methods and approach used are cool, I found the way the authors treated “trichromacy” and defined “trichromatic species” to be very problematic and not well explained. In the Introduction, they present a simplified view of primate trichromacy as primarily occurring in the Old World monkeys and apes (line 71), ignoring the widespread presence of polymorphic trichromacy in primates in the Americas and in several day-active species in Madagascar. Polymorphic trichromacy is where some females in a species are trichromats while all males and remaining females are dichromats (due to allelic variation at the LWS opsin gene). The proportion of individuals in a species that are trichromats can vary depending in part on the number of LWS opsin alleles that are present in the species.

While it is possible that different selective pressures shaped the evolution of polymorphic trichromacy compared to the “routine trichromacy” of the Old World species, the authors do not offer any explanation as to why they exclude polymorphic species as “trichromats” in their analyses. This is problematic, because many of the studies in the literature that have discussed the benefit of trichromacy for detecting conspicuous foods, like fruits, have been performed on New World polymorphic species. Two of these previous studies (e.g., Regan et al. 2001; Meliln et al. 2017) are in fact cited by the authors of this manuscript in the Introduction.

Because of this oversight in how they define “trichromat species”, I had trouble trusting their measures of “trichromat richness” and subsequent results. I do appreciate that they discuss polymorphic species in some parts of the Results, such as when they report that the proportion of conspicuous fruits was correlated with polymorphic species richness, although to a weaker extent. I wonder whether it could be beneficial to reframe the manuscript to emphasize that there are different types of primate trichromacy, and there is an effect of conspicuous fruits on both types of trichromacy (although weaker for polymorphic) – e.g., analyze both types separately and together for all of the methods? It just felt disingenuous in the Introduction to not mention polymorphic trichromacy at all, despite its widespread prevalence.

Thank you very much for pointing out this concern, and providing a good solution. We followed the reviewer's advice and now present the manuscript by looking at polymorphic, trichromatic and polymorphic + trichromatic primates (combined), and present these results in the main text (Fig. 2) and in the supplementary information (mainly Fig. S2). In addition, we repeated all sensitivity analyses for polymorphic primates as well (see Table S2). We rephrased the introduction to further introduce these different colour vision systems:

L76: “Interestingly, primates show variations in trichromatic colour vision systems. Some species have ‘polymorphic’ colour vision (primarily New World monkeys), in which part of the population is dichromatic and part is trichromatic, whereas other species are ‘routine trichromatic’ (primarily Old World monkeys and apes) in which all individuals have trichromatic vision. This variation may explain why the ecological factors that have driven

the evolution of colour vision systems in primates remain debated (Regan et al., 2001; Valenta et al., 2018)."

Interestingly, although results remain very similar when including polymorphic primates as well, these results are not robust when we compare the observed pattern to results from simulations in which we randomly reshuffle primate species across botanical countries, and repeat the analyses. This means that polymorphic vision (which preceded routine trichromacy in an evolutionary sense) may have evolved in response to different selection pressures than fruit colours, or at least several selection pressures may have been at play to maintain the variation of colour vision systems in these species, and determine the present-day distribution of these species. Routine trichromacy remains overwhelmingly associated with conspicuous palm colours in all analyses, and is further supported in the grid-based analyses performed on the African mainland (where no polymorphic primates occur).

If the authors only want to test effects of conspicuous fruits on the species richness of routine trichromats, they should specify that and justify the decision. They should also acknowledge that routine trichromats are phylogenetically constrained to Africa and Asia (but are not found in Madagascar), except for the convergent evolution of routine trichromacy in one genus of New World monkey (*Alouatta*). I am unfamiliar with structural equation modeling – do you also need to take phylogenetic history into account when determining trichromat species richness?

As indicated above, we followed the suggestion to perform all analyses also including polymorphic primates. For SEM analyses it is currently not possible to correct for phylogenetic history, but by performing simulations we can at least compare our results to a random distribution of primates (keeping total numbers of primate species and colour vision the same) as also pointed out in our response to reviewer #1. We have now also emphasized that colour vision systems are strongly constrained (phylogenetically and geographically) and that this may have an impact on our results:

L236: "Our results show that at a global scale, the geographical distribution of trichromatic primates is associated with the availability of conspicuous relative to cryptic palm fruits (Figure 2). However, when including polymorphic primates, our results do not deviate from a neutral expectation in which primates are randomly distributed across regions. This may be explained by the fact that colour vision systems evolved only few times independently within primates (probably < 7 times, Figure S11), and colour vision is thus geographically (Figure 2) and phylogenetically (Figure S11) constrained. Thus, our results strongly support the fruit foraging hypothesis (Allen, 1879) for routine trichromacy, especially in Africa (Figure 2, Figs. S7, S8), but the ecological determinants of polymorphic vision remain questionable."

Other concerns:

1. Primate species color vision classification needs to be updated. All *Eulemur* species except *E. flavifrons* and *E. macaco* are dichromats (Jacobs et al. 2019).

Thank you, we have updated the data (Appendix S1) and included this in the new analyses.

2. One of the major hypotheses proposed to explain routine trichromacy is the exploitation of protein-rich young red leaves by catarrhine primates during periods of food scarcity (as a fallback resource). I felt that this hypothesis was not given sufficient attention. I was also curious as to whether the authors observed an increase in diurnal folivorous trichromat species from 10 Ma onward. They mention that there is an increase in diurnal frugivorous trichromats but not in non-trichromat species. The colobine subfamily of Cercopithecidae primates are all routine trichromats and have all evolved specialized stomachs for folivory. If there was an increase in both frugivorous and folivorous trichromats, it might be difficult to link the increase in species richness specifically with fruit coloration rather than an adaptive

radiation of cercopithecids in general during the last 10 Ma.

Thank you for raising this point. We have redone the analysis of ancestral diversity changes through time by looking at polymorphic/trichromatic/dichromatic primates globally and in the different geographical realms. In addition, we have looked at diversity changes through time for frugivores vs. folivores in Africa (mainland). Looking at these traits (colour vision and diet) simultaneously is difficult, because they may have at least partly evolved independently from each other. Our results indeed show that the fast diversification of trichromats on the African mainland may be partly linked to frugivores but also largely includes folivores. It is therefore difficult to disentangle the difference between frugivores and folivores in a temporal, evolutionary context. We have included this in the discussion, and emphasized that it remains possible that trichromatic vision in Africa is linked to both frugivory and folivory, as well as the other primate traits that co-evolved during this time. See Fig. S15 for these new reconstructions of folivores vs. frugivores.

Discussion:

L245 (and onwards): “Trichromatic colour vision in Africa has also been hypothesized to be adaptive to detecting young, reddish leaves (Dominy et al., 2003). Domini et al. (Dominy et al., 2003) argued that increased seasonality after the onset of the Oligocene-Miocene led to an overall reduction in keystone plant species, including palms (Pan et al., 2006; Blach-Overgaard et al., 2013), and an increase in periods of fruit scarcity. A fallback diet of young, reddish leaves may thus have been the selective pressure for the evolution of routine trichromatic vision (from a polymorphic ancestor, Figure S11) in these catarrhine primates. Although we cannot exclude this possibility, and detailed spatial and phylogenetic data on leaf coloration may allow the testing of these competing hypotheses in future research, results from our sensitivity analyses suggest that trichromacy richness is not simply correlated to conspicuous palm fruits due to confounding correlations between fruits and other variables (Table S2).”

And:

L267: “The African trichromatic radiation includes primarily frugivorous species, and no difference in diversification of folivorous (leaf-eating) compared with non-folivorous species was detected (Figure S15).”

References:

- Allen G. 1879.** *The colour-sense: its origin and development: an essay in comparative psychology (Vol. 10)*. London: Trübner & Co.
- Bentley-Condit VK. 2009.** Food choices and habitat use by the Tana River yellow baboons (*Papio cynocephalus*): a preliminary report on five years of data. *American Journal of Primatology* **71**(5): 432-436.
- Blach-Overgaard A, Kissling WD, Dransfield J, Balslev H, Svenning J-C. 2013.** Multimillion-year climatic effects on palm species diversity in Africa. *Ecology* **94**(11): 2426-2435.
- Chapman C, Russo S 2007.** Primate seed dispersal. Linking behavioral ecology with forest community structure. In: Campbell C, Fuentes A, Mackinnon K, Panger M, Bearder S eds. *Primates in perspective*. Oxford: Oxford University Press, 510–525.
- Dominy NJ, Svenning J-C, Li W-H. 2003.** Historical contingency in the evolution of primate color vision. *Journal of Human Evolution* **44**(1): 25-45.
- Fleming TH, Kress WJ. 2013.** *The ornaments of life: coevolution and conservation in the tropics*: University of Chicago Press.
- Homewood KM. 1978.** Feeding strategy of the Tana mangabey (*Cercocebus galeritus galeritus*) (Mammalia: Primates). *Journal of Zoology* **186**(3): 375-391.
- Kulp J, Heymann EW. 2015.** Ranging, activity budget, and diet composition of red titi monkeys (*Callicebus cupreus*) in primary forest and forest edge. *Primates* **56**(3): 273-278.
- Palacios E, Rodríguez A, Defler TR. 1997.** Diet of a group of *Callicebus torquatus lugens* (Humboldt, 1812) during the annual resource bottleneck in Amazonian Colombia. *International Journal of Primatology* **18**(4): 503-522.
- Pan AD, Jacobs BF, Dransfield J, Baker WJ. 2006.** The fossil history of palms (Arecaceae) in Africa and new records from the Late Oligocene (28–27 Mya) of north-western Ethiopia. *Botanical Journal of the Linnean Society* **151**(1): 69-81.
- Regan BC, Julliot C, Simmen B, Viénot F, Charles–Dominique P, Mollon JD. 2001.** Fruits, foliage and the evolution of primate colour vision. *Philosophical Transactions of the Royal Society of London. Series B: Biological Sciences* **356**(1407): 229-283.
- Santhosh K, Kumara HN, Velankar AD, Sinha A. 2015.** Ranging behavior and resource use by lion-tailed macaques (*Macaca silenus*) in selectively logged forests. *International Journal of Primatology* **36**(2): 288-310.
- Schreier AL. 2010.** Feeding ecology, food availability and ranging patterns of wild hamadryas baboons at Filoha. *Folia Primatologica* **81**(3): 129-145.
- Tutin CEG, Ham RM, White LJT, Harrison MJS. 1997.** The primate community of the Lopé reserve, Gabon: Diets, responses to fruit scarcity, and effects on biomass. *American Journal of Primatology* **42**(1): 1-24.
- Valenta K, Nevo O, Chapman CA. 2018.** Primate fruit color: Useful concept or alluring myth? *International Journal of Primatology* **39**(3): 321–337.
- Zona S, Henderson A. 1989.** A review of animal-mediated seed dispersal of palms. *Selbyana* **11**: 6-21.

Appendix B

Associate Editor Board Member

Comments to Author:

Two referees have now seen the revised version of your manuscript and as you will see both of them were very pleased with the efforts you made to address their previous comments and criticisms. Moreover, both of them feel that your contribution is important and interesting. Nonetheless both have a number of suggestions for further improvement. In particular, Reviewer 1 still has issues with some of your conclusions not being supported by the data (including one of the main conclusions) and also feels that you under-sell yourselves when presenting other data that have very interesting implications for the evolution of colour vision. I thus recommend that you carefully consider these comments and suggestions and adjust your text accordingly.

We thank the Associate Editor and two referees for their positive feedback and the constructive comments, which were fully taken into account in the revision. In our response to the referees, we reply to all comments in detail (in blue) and indicate where and how we have dealt with them (the changed text parts are indicated with tracked changes in the manuscript). Specifically, we removed the conclusion suggesting that trichromacy has been adaptive to frugivory, because we do not have sufficient independent events to make this claim (reviewer #1). Furthermore, we followed the suggestion to emphasize how trichromats have also shaped palm radiations of species with conspicuous fruits in Africa (reviewer #1). We also restructured the manuscript (reviewer #1) and rephrased some of the sentences (reviewer #2). We think that the manuscript has further improved with the comments from the referees and we hope you will agree.

Reviewer(s)' Comments to Author:

Referee: 1

Comments to the Author(s).

The manuscript is much improved and that the new analyses contribute significant findings to the previous research. This is an exciting dataset and I feel the authors have uncovered several interesting results. However, I still feel that not all of the conclusions (including a main conclusion) are justified by the results, and I think the authors can and should highlight discoveries they have made that they are not currently focusing on.

Thanks very much for these additional suggestions!

The first conclusion in the last sentence of the abstract is supported by the results (co-diversification of palms and trichromats) but the second claim is not justified by the results. Given that genetic evidence shows a single origin of trichromacy in catarrhines (the diurnal primates currently on mainland Africa) co-diversification of palms across Africa happened once routine trichromacy was established – Aspects shaping the radiation of primate species after this event cannot explain the evolution of trichromacy.

This is true and we have removed (or rephrased) the last sentence of the abstract, the introduction, as well as a sentence from the conclusion related to this.

The authors do have evidence that trichromats might have selected for radiation of colorful (vs cryptic) palms – and this would be a very interesting and exciting result that should be given more spotlight. That similar patterns are not found in polymorphic species suggests to be those primates don't shape fruit traits in the same way – i.e. the evolutionary relationships between primates and palms are impacted by primate color vision.

We followed the suggestion to emphasize this result in the discussion and the last sentence of the abstract:

L47: *“These results suggest that the distribution and diversification of African trichromatic primates is strongly linked to the relative availability of conspicuous (vs. cryptic) palm fruits, and that interactions between primates and palms are impacted by the co-evolutionary dynamics of primate colour vision systems and palm fruit colours.”*

L315: *“Furthermore, our results suggest that trichromats have also influenced the radiation of conspicuous (vs. cryptic) palms. The absence of a relationship between the diversification of palms and polymorphic primates suggests that polymorphs may not shape fruit colours in the same way as trichromats do, and fruit traits other than colour (e.g. husk thickness, fruit size, fruit odor, nutritional content and the presence of juicy soft pulp) may play a more important role in the selection of fruits by polymorphs.”*

Overall - the original selective pressure for trichromacy is not clarified by the present analysis. If the authors could show conspicuous palms were more common in areas where first catarrhines evolved but not where the polymorphic trichromats evolved this would be more direct evidence. I'm not suggesting this is possible, just giving an example that I hope clarifies my point. Overall - I feel the paper needs reworking throughout to clarify what is being tested and what can be concluded. But to be clear – I think the analyses and results are comprehensive, novel and exciting – I feel they are just making a slightly different contribution that the present focus of the manuscript.

We hope that this revision clarifies what can be concluded based on the results.

Minor points –

Use “palms” or “palm fruits” and not just “fruits” throughout. It is misleading to equate “palm” with “fruit” – this is done throughout the MS and should be changed, or it will lead to misquoting. (I agree palms are important, but they are a small fraction of total fruits).

We changed this throughout the manuscript into ‘palm fruits’, including in the title.

Line 67 – “frugivory hypothesis” and not “foraging” hypothesis. Leaf eating is also “foraging” and a distinct hypothesis.

Thanks, we changed this.

Line 103 – 115 The three “hypotheses” given by the authors are all predictions stemming from a single hypothesis that primate color vision and fruit traits are related.

Yes, this is true. We now defined the main hypothesis (that the origin, distribution and diversity of trichromatic vision in primates is positively associated with the availability of conspicuous fruits of palms (Arecaceae)) and then present the specific expectations as predictions.

Please organize the results and discussion into a structure that follows the predictions (at present 3 predictions/ hypotheses but there are 4 sections (a-d) in the results and 2 sections (a-b) in the discussion. This disrupts the flow of the paper and makes it hard to follow.

This is a great suggestion and we have now followed the order of the four predictions throughout the results and the discussion.

Line 165 - How are cathemeral and crepuscular primate species handled in your diurnal vs

nocturnal scenario? This is an important consideration, especially considering the variation between and between species and regions in Madagascar (some lemurs are nocturnal at some sights, and cathemeral at others, for example). The methods for assigning primate activity patterns should be acknowledged clearly and stated how the authors assigned categories.

For activity status, we collected data on whether a species is day-active or not (i.e. night-active or crepuscular) from the EltonTraits 1.0 database (317 primate species, Wilman et al., 2014), supplemented with data from the Handbook of the Mammals of the World: Primates (61 species, Mittermeier et al., 2013). For 27 primate species the activity data were interpolated from the genus. These species belong to eight genera, and in all of these genera the activity level is conserved (see Appendix S1).

This information is provided in the Supplementary Information. For some of the analyses we focused on day-active primates only, in which case we only extracted those primates that are day-active. Non-day-active species thus include night-active, crepuscular and cathemeral species.

The discussion is overlong and could be usefully shortened.

We have now shortened it, partly by the restructuring following the four predictions, and also partly by removing some of the examples.

I'd be happy to look at another version of the manuscript.

Best wishes,

Amanda Melin

Referee: 3

Comments to the Author(s).

In this manuscript, Onstein and colleagues use novel methods to explore a longstanding question in mammal vision – what factors have influenced the evolution of trichromatic color vision in primate clades? I reviewed a previous version of this manuscript, and it is very clear that the authors took the reviewer comments to heart and made substantive changes to the manuscript that have greatly improved it. I think this manuscript now offers an exciting and novel contribution to the research area and employs really cool methodology.

I have only minor suggestions for revisions. Primarily, these involve citing statements more thoroughly, clarifying some sentences, and fixing a typo.

Thanks very much for these additional suggestions!

- Lines 77-80: you should include citations or at least a citation for the types/distribution of color vision. Gerald Jacobs has great reviews you could cite.

We added Jacobs et al. 1993.

- Lines 213-215: how did you estimate the dates for the evolution of polymorphic trichromacy and routine trichromacy? I think you should include the citation for the phylogeny you use in the main text and not just supplemental.

This is described in the methods section; in which we also include the reference to the phylogeny.

L367: *“To obtain a phylogeny for extant primates, we pruned the mammal phylogenetic tree from [48] to include our taxa of interest (n = 385 primate species, i.e. 26 primate species were missing from the phylogeny)”*

And L484: *“To assess whether evolutionary radiations of trichromatic and/or polymorphic primates and palms with conspicuous fruits are synchronized (P4), we visualized the evolution of trichromatic and polymorphic colour vision and conspicuous palm fruits on the primate and palm phylogenetic trees, respectively (Figure S11).”*

- Lines 251-254: *“Although we cannot exclude this possibility, and detailed spatial and phylogenetic data on leaf coloration may allow the testing of these competing hypotheses in the future, results from our sensitivity analyses ...”* This sentence is fairly convoluted and it's hard to follow. It may work better if you break it up or expand it.

Thanks, we rephrased:

L280: *“To test this competing hypothesis, we could use a similar SEM approach as done here for palm fruit colours, but then with detailed spatial and phylogenetic data of leaf coloration. These data are currently unavailable. However, results from our sensitivity analyses suggest that trichromacy richness is not simply correlated to conspicuous palm fruits due to confounding correlations between fruits and other variables, such as reddish leaves (Table S2).”*

Similarly, I think you should expand what you mean regarding the new sentence from 267-269.

We also rephrased:

L310: *“Indeed, although diffuse co-evolution between primates and angiosperms may date back to the Early Cenozoic (ca. 66 Ma) [31], our results suggest that, at least in Africa, mutualism-dependent diversification between frugivorous trichromatic primates and conspicuous palms may have happened from c. 10 Ma onward (Middle to Late Miocene) (Figure 4). This contrasts with folivorous primates which did not diversify faster or slower than non-folivorous primates in Africa (Figure S15).”*

- Line 264: lowercase “A” for Although, since the sentence begins “Indeed, although ...”

Done.

Appendix C

We are delighted to submit our revised manuscript entitled “**Palm fruit colours are linked with the broad-scale distribution and diversification of primate colour vision systems**” by Renske E. Onstein, Daphne N. Vink, Jorin Veen, Christopher D. Barratt, Suzette G.A. Flantua, Serge A. Wich and W. Daniel Kissling for consideration as an *Article in Proceedings of the Royal Society B*.

We thank the Associate Editor and referee for their additional minor comments and suggested textual changes, which were all fully taken into account in the revision. In our response to the referee, we reply to all comments in detail (in blue) and indicate where and how we have dealt with them.

Yours sincerely, on behalf of all authors,
Renske Onstein

Reviewer(s)' Comments to Author:

Referee: 1

Comments to the Author(s)

The current title maintains adherence to the previous “causation” framing, which the authors agreed was flawed, and should be revised to something like: Palm fruit colors are correlated with the broad-scale distribution and diversification of primate color vision phenotypes.

The term ‘determine’ is commonly used in macroecological literature to refer to statistical relationships among variables (i.e. correlations), but we follow the reviewer and have changed the title to: “Palm fruit colours are linked with the broad-scale distribution and diversification of primate colour vision systems”

Minor suggestions are included as tracked changes in the word document.

Thank you very much, we accepted all textual suggestions and also changed several sentences in the abstract and introduction, as suggested.

Supplementary Methods (No line numbers were provided, so hopefully these edits are easy to locate)

Page 1

Given some of these data are interpolated from higher taxonomic levels, revise the opening sentences to: We collected or inferred/ interpolated functional trait data from.....

Done.

I suggest you remove the following as you don't focus on the tuning of pigments: “Although the precision of the collected vision data was much higher at the species level compared to the family level or the interpolated level, distinct primate lineages have a conservative spectral tuning of their photopigments (SurrIDGE et al., 2003). “

Done, we removed this.

Page 1-2

Of the fruit species you classify as “cryptic” not only the purple, but also white and black and blue and essentially all non-green colors would likely be discriminable by trichromats and dichromats due to luminance difference. “Cryptic” (suggestive of color-matching with leaves) is therefore not the best name. I suggest you call them “Other” or “Non-conspicuous” in which this category includes greenish cryptic fruits. So, you would have 2 categories: “Conspicuous to trichromats (red, orange, yellow, conspicuous for short)” and “Other” (Green, blue, purple, black, white, ivory etc etc). Please revised accordingly here and in main text and Appendix S2.

We followed the reviewer's suggestion and changed all use of ‘cryptic’ into ‘non-conspicuous’ (in main text, SI, appendices and figures).

Suggest avoiding colonial terms of “New World/ Neotropical” and “Old World/ Paleotropical” and describe them as monkeys of the Americas or African and Asian Primates etc.

We changed this throughout the SI and main text using Americas/Africa/Asia instead.

The revisions to remove the previous framing around testing the frugivory hypothesis were not entirely thorough, leaving the flow of the introduction a bit disjointed. The authors should at minimum add in their introduction a few sentences in opening paragraph(s) concerning the potential for dispersers to shape fruit communities and remove continued adherence to old framing – a few listed below:

We added the following sentence to the first paragraph to introduce this:

“Frugivores and fruit plants are therefore able to shape each other’s traits and the composition and structure of ecological communities [7].”

Title (see above): Suggestion: Palm fruit colors are correlated with the broad-scale distribution and diversification of primate color vision phenotypes.

See above, we changed the title to account for this suggestion.

Line 312 – are consistent with, is more appropriate than “strongly support”

Done, changed as suggested.

End of Supplementary Methods: “In conclusion, the results including figs indicate that figs, as compared to palms, probably played a less important role for trichromat distribution and diversification in mainland Africa (also see Dominy et al., 2003).” (Please revise)

It’s not entirely sure to us why this sentence needs to be changed? But assuming it’s simply unclear, we rephrased to: “In conclusion, the results including figs indicate that fig fruits probably played a less important role for trichromat distribution and diversification in mainland Africa than palm fruits (also see Dominy et al., 2003)”

The opening sentence of abstract – you don’t test the hypothesis that you start with in your abstract. I suggest you are consistent with the hypothesis you adopt in the text of your revised paper and quote in the response to reviewers: “that the origin, distribution and diversity of trichromatic vision in primates is positively associated with the availability of conspicuous fruits of palms (Arecaceae)”

We changed the abstract to clarify this hypothesis:

“Here, we test the hypothesis that the origin, distribution and diversity of trichromatic primates is positively associated with the availability of conspicuous palm fruits, i.e. keystone fruit resources for tropical frugivores.”